# DirMoE: Dirichlet-Routed Mixture of Experts

**Amirhossein Vahidi**[1,*]     **Hesam Asadollahzadeh**[1,2,*]     **Navid Akhavan Attar**[2]
**Marie Moullet**[1]     **Kevin Ly**[1]     **Xingyi Yang**[3]     **Mohammad Lotfollahi**[1]

[1]Wellcome Sanger Institute, Wellcome Genome Campus, Cambridge, UK
[2]School of Computing and Information Systems (CIS), Faculty of Engineering and IT (FEIT),
 University of Melbourne, Australia
[3]The Hong Kong Polytechnic University, Hong Kong
{ha11,av13,ml19}@sanger.ac.uk
*Equal contribution

## Abstract

Mixture-of-Experts (MoE) models have demonstrated exceptional performance in large-scale language models. Existing routers typically rely on non-differentiable Top-$k$+Softmax, limiting their performance and scalability. We argue that two distinct decisions, which experts to activate and how to distribute expert contributions among them, are conflated in standard Top-$k$+Softmax. We introduce Dirichlet-Routed MoE (DirMoE), a novel end-to-end differentiable routing mechanism built on a Dirichlet variational autoencoder framework. This design fundamentally disentangles the core routing problems: expert selection, modeled by a Bernoulli component, and expert contribution among chosen experts, handled by a Dirichlet component. The entire forward pass remains fully differentiable through the use of Gumbel-Sigmoid relaxation for the expert selection and implicit reparameterization for the Dirichlet distribution. Our training objective, a variational ELBO, includes a direct sparsity penalty that precisely controls the number of active experts in expectation, alongside a schedule for key hyperparameters that guides the model from an exploratory to a definitive routing state. Moreover, our DirMoE router matches or exceeds other methods while improving expert specialization.

## 1 Introduction

Modern sparse Mixture-of-Experts (MoE) layers scale capacity without proportional computation by routing each token to a small subset of experts (Shazeer et al., 2017; Jiang et al., 2024; DeepSeek-AI, 2024; Du et al., 2021). The main component of MoE is the router. The router is responsible for answering two questions: (i) which experts participate and (ii) their degree of contribution, determined by the probability (mass) that is distributed among them. Furthermore, the learning process should be sparse, with only a fraction of active experts, in order to decrease computational complexity. One of the most prominent approaches to address this is using Top-$k$+Softmax. A persistent challenge in standard Top-$k$+Softmax routing (Lepikhin et al., 2020; Fedus et al., 2022) is the lack of end-to-end gradients through the discrete selection step and its requirement for a secondary objective (e.g., temperature tuning, auxiliary losses, straight-through estimators) to encourage sparsity during training, which complicates stability and calibration. Beyond optimization, the single Softmax entangles two decisions: expert selection and how much their contribution is, making expert usage hard to interpret. This entanglement ties load balancing to mixture calibration via a single temperature, relying on capacity constraints to curb overload, which causes an uneven distribution across experts. These limitations motivate a routing mechanism that (i) preserves full differentiability and (ii) offers explicit, interpretable control over expert selection and probability allocation on the simplex.

Assignment-based routers (BASE Layers) reframe dispatch as an assignment problem to guarantee balanced loads (Lewis et al., 2021); this addresses balance but not mixture calibration or interpretability of how much mass the active set receives. Most recently, ReMoE replaces Top-$k$+Softmax with a fully continuous gate to restore end-to-end differentiability and improve scalability (Wang et al., 2025; Puigcerver et al., 2024). While this directly tackles the gradient bottleneck,

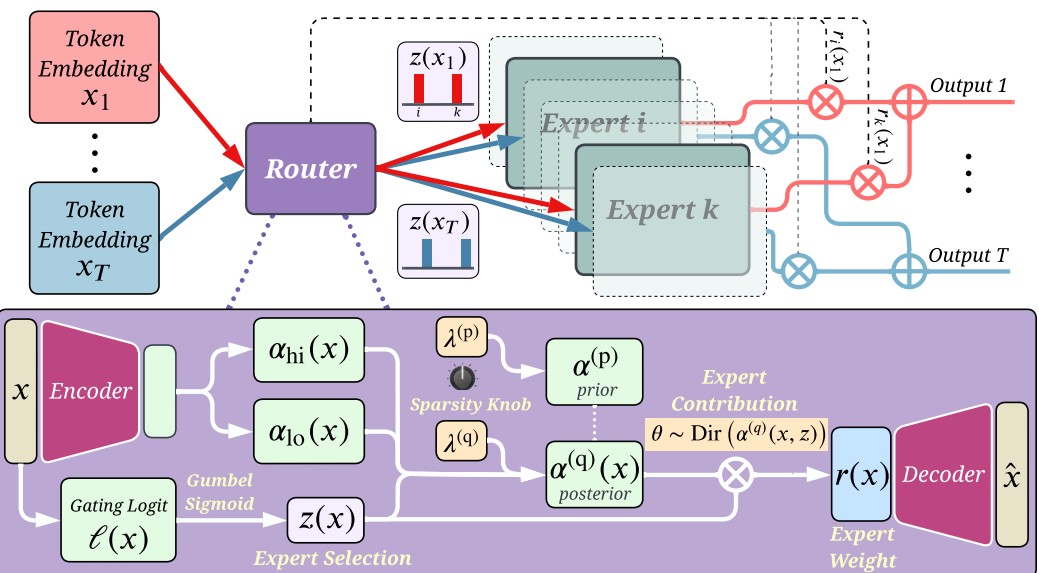

Figure 1: Illustration of **DirMoE**. Given a batch $\mathcal{X}$ of input embeddings, two different heads $\alpha_{\mathrm{hi}}(x), \alpha_{\mathrm{lo}}(x)$ learns the active and inactive per-token expert concentration, and $\ell(x)$ learns the gating logits. The routing probabilities are the normalized product of $z$ expert selection and Dirichlet probabilities $\theta$ (expert contribution).

the resulting "soft" routing needs an auxiliary loss to enforce sparsity. In practice, these can inject interference gradients, complicate tuning, and dampen expert specialization (Wang et al., 2024). Moreover, when they are combined with capacity-based routing (token dropping), they may introduce instability (Rajbhandari et al., 2022; Gale et al., 2023).

We introduce a **Dirichlet variational router** that factorizes routing into (i) a binary selection vector over experts (the set of active experts) and (ii) a Dirichlet distribution over mixture weights on the simplex (the distribution among the active experts). As shown in Figure 1, the router produces a relaxed expert selection vector via a Gumbel–Sigmoid reparameterization and the expert contribution via a Dirichlet; the final routing weights are the normalized Hadamard product of the two components (see Section 4). Furthermore, the router is fully differentiable with implicit reparameterization (Maddison et al., 2017; Jang et al., 2017; Figurnov et al., 2018). A key benefit is calibration. Under a Dirichlet prior, the total mass assigned to any subset of experts follows a Beta distribution. We exploit this to derive a one-parameter (sparsity knob) control that sets the expected mass on the active set, decoupled from the size of active experts ($k$). Thus, $k$ cleanly determines how many experts participate, while a Beta-calibrated ratio of Dirichlet shape parameters determines how much they collectively contribute. This design retains the end-to-end gradient advantage and adds a transparent sparsity knob to calibrate *expert selection* and *expert contribution*.

**Our contributions are:**

- We introduce a novel probabilistic router to improve sparsity in MoE. **DirMoE** is the first method using a Dirichlet variational autoencoder producing sparse, interpretable routes on the simplex for MoE.

- We use Gumbel-sigmoid for expert selection and implicit reparameterization for Dirichlet samples to enable end-to-end gradients. This design maintains the MoE's full differentiability.

- We introduce a coefficient (sparsity knob) that can control the sparsity of the router, providing a practical calibration knob for specialization and sparsity.

- We demonstrate the strong performance of our method across multiple datasets and tasks: **DirMoE** shows competitive zero-shot results and improves expert specialization.

## 2 RELATED WORK

**Mixture-of-Experts and Routing** GShard and Switch established practical routing via auxiliary balancing and capacity control (Lepikhin et al., 2020; Fedus et al., 2022); GLaM demonstrated trillion-parameter efficiency (Du et al., 2021). To avoid discrete Top-$k$, recent differentiable routers restore end-to-end gradients: Soft-MoE replaces hard selection with soft assignments (Puigcerver et al., 2024); ReMoE uses ReLU routing with explicit sparsity control and balancing (Wang et al., 2025); and Lory scales fully differentiable MoE to pretraining via segment routing and similarity-based batching (Zhong et al., 2024). To address the limitation of independent routing at each layer, Layerwise Recurrent Router (RMoE) introduces a GRU-based mechanism to propagate routing states, enabling cross-layer dependencies for improved expert selection (Qiu et al., 2025).

**Sparsity in MoE** A central goal in MoE is to achieve controllable sparsity between different experts. Controllable sparsity has evolved from temperature and capacity heuristics and balancing losses (Fedus et al., 2022; Zoph et al., 2022) toward learned or auto-tuned activation budgets. Variable-$k$ routing allows each token to choose its expert count: DynMoE jointly learns per-token $k$ and the global number of experts (Guo et al., 2025), while DA-MoE derives token importance from attention to allocate a variable number of experts (Akhavan Aghdam et al., 2024). In contrast, dynamic depth allocates compute across layers rather than experts: Mixture-of-Depths (MoD) learns to skip or process tokens under a compute budget (Raposo et al., 2024), A-MoD routes using attention maps without extra parameters (Gadhikar et al., 2024), and Router-Tuning (MindSkip) enables dynamic depth by training only a lightweight router (He et al., 2024).

## 3 PRELIMINARY

### 3.1 SPIKE AND SLAB ROUTING ON THE SIMPLEX

We model MoE routing with a spike and slab prior (Mitchell & Beauchamp, 1988; Ishwaran & Rao, 2005) that separates which experts are active (expert selection) from how probability mass is distributed among the active experts (expert contribution). Let $E$ be the total number of experts, $\mathbf{z} \in \{0, 1\}^E$ be a binary selection mask and $\boldsymbol{\theta} \in \Delta^{E-1}$ a simplex vector. We factorize

$$p(\mathbf{z}, \boldsymbol{\theta} \mid x) \;=\; \underbrace{\prod_{i=1}^{E} \mathrm{Bernoulli}(z_i \mid \pi_i(x))}_{\text{spike: Expert selection}} \;\times\; \underbrace{\mathrm{Dir}\big(\boldsymbol{\theta} \mid \boldsymbol{\alpha}^{(p)}(\mathbf{z})\big)}_{\text{slab: Expert contribution}}, \tag{1}$$

where $\pi_i(x) \in (0, 1)$ are gate priors (optionally with Beta hyperpriors) and $x$ is the token embedding input to the MoE layer. The slab uses a two-level concentration:

$$\alpha_i^{(p)}(\mathbf{z}) \;=\; \lambda\Big(z_i\,\alpha_{\mathrm{hi}} + (1 - z_i)\,\alpha_{\mathrm{lo}}\Big), \qquad \alpha_{\mathrm{hi}} > \alpha_{\mathrm{lo}} > 0, \;\; \lambda > 0. \tag{2}$$

Where $\alpha_{\mathrm{hi}}$ is the per-expert concentration when an expert is active ($z_i \approx 1$), $\alpha_{\mathrm{lo}}$ is the per-expert concentration when inactive ($z_i \approx 0$); lower number reduces leakage into inactive experts. $\lambda$ is a global scale of the concentration of active and inactive experts, it controls dispersion in the Dirichlet parameter. A larger $\lambda$ tightens samples around the mean so the distribution becomes more uniform. A smaller $\lambda$ yields higher variance and more mass near simplex vertices, so the distribution is more spiky (effectively sparser draws).

### 3.2 SIMPSON INDEX AS A SPARSITY METRIC

In the context of probability distributions, the Simpson Index is defined as the sum of the squared probabilities: $H(\mathbf{p}) = \sum_{i=1}^{E} p_i^2$. This metric provides an intuitive measure of a distribution's concentration: a perfectly sparse distribution with all probability mass on one outcome provides an upper bound of $H = 1$, while a uniform distribution across all $E$ outcomes yields a lower bound of $H = 1/E$. For any other distribution, the value lies between these two bounds. We use the Simpson index as a direct and interpretable measure of sparsity for our routing distributions, where a higher value indicates greater sparsity.

## 4 METHOD

In the following, we introduce **DirMoE**, a fully differentiable MoE router that factors routing into two components: expert selection and expert contribution, and how they define the final expert weights. Then we introduce a new parameter $\lambda$, the sparsity knob that can control sparsity.

### 4.1 PROBLEM SETUP

We consider a transformer decoder with sparse MoE layers. Given a token embedding $x \in \mathbb{R}^d$, let $\{E_i\}_{i=1}^E$ denote expert networks and let a router produce logits $\ell(x) \in \mathbb{R}^E$. We seek a routing vector $\mathbf{r}(x) \in \Delta^{E-1}$ that satisfies: (i) only $k \ll E$ experts are active per token; (ii) intrinsic activation sparsity for interpretability; and (iii) full differentiability so the router can learn task-dependent sparsity.

### 4.2 DIFFERENTIABLE ROUTER

The router has two components: (i) expert selection generated from logits $\ell(x) \in \mathbb{R}^E$; combined below. The selection indicator is trained via binary Gumbel-Sigmoid Jang et al. (2017); Maddison et al. (2017). Given router logits $\ell(x)$, we sample an expert selection vector:

$$\tilde{z}_i = \sigma\left(\frac{\ell_i(x) + g_i}{\tau_z}\right), \quad g_i \sim \text{Logistic}(0,1), \tag{3}$$

where $\sigma$ is the sigmoid function, $\tau_z > 0$ is the temperature. This yields a soft mask $\tilde{z} \in (0,1)^E$; as $\tau_z \downarrow 0$, $\tilde{z}$ becomes nearly binary.

(ii) Conditioned on $\tilde{\mathbf{z}}$, we define a Dirichlet variational posterior over a latent simplex variable $\boldsymbol{\theta}$,

$$q_\phi(\boldsymbol{\theta} \mid x, \tilde{\mathbf{z}}) = \text{Dir}\big(\boldsymbol{\alpha}^{(q)}(x, \tilde{\mathbf{z}})\big),$$

with component-wise parameters

$$\alpha_i^{(q)}(x, \tilde{\mathbf{z}}) = \lambda^{(q)}\Big(\tilde{z}_i \, \alpha_{\text{hi},i}(x) + (1 - \tilde{z}_i) \, \alpha_{\text{lo},i}(x)\Big), \tag{4}$$

where $\lambda^{(q)} > 0$ scales the total concentration and $\alpha_{\text{hi},i}(x), \alpha_{\text{lo},i}(x) > 0$ may be learned functions of $x$ (or shared across $i$) and the Dirichlet is sampled via normalized Gamma with implicit reparameterization (Figurnov et al., 2018).

For the prior, we use the same relaxed gates to avoid mixing discrete and continuous objects and to keep a closed-form KL:

$$p(\boldsymbol{\theta} \mid \tilde{\mathbf{z}}) = \text{Dir}\big(\boldsymbol{\alpha}^{(p)}(\tilde{\mathbf{z}})\big), \qquad \alpha_i^{(p)}(\tilde{\mathbf{z}}) = \lambda^{(p)}\Big(\tilde{z}_i \, \alpha_{\text{hi}} + (1 - \tilde{z}_i) \, \alpha_{\text{lo}}\Big), \tag{5}$$

with constants $\alpha_{\text{hi}} > \alpha_{\text{lo}} > 0$ and scale $\lambda^{(p)} > 0$. This is a smooth surrogate to a spike and slab on faces of the simplex: as $\tilde{z}_i \to 0$, coordinate $i$ is driven toward (near-)zero probability; as $\tilde{z}_i \to 1$, it joins the active slab.

We introduce the router as a normalized combination of two vectors (i) the expert selection, and (ii) the expert contribution $\theta$ as follows:

$$\boxed{\mathbf{r}(x) = \text{normalize}\big(\tilde{\mathbf{z}}(x) \odot \boldsymbol{\theta}(x)\big) \in \Delta^{E-1}, \quad \boldsymbol{\theta} \sim \text{Dir}\big(\boldsymbol{\alpha}^{(q)}(x, \tilde{\mathbf{z}})\big)} \tag{6}$$

which gates the experts' outputs. Our goal is to have $\|\tilde{z}(x)\|_0 \approx k$ and to place almost all of the total probability mass of $\alpha^{(q)}$ on the active set $S(x) = \{i : z_i(x) = 1\}$. Then the MoE output is :

$$y(x) = \sum_{i=1}^E r_i(x) \, E_i(x). \tag{7}$$

This design maintains full differentiability throughout the entire forward pass: gradients flow through the Binary-Concrete relaxation (Maddison et al., 2017; Jang et al., 2017) and through the Dirichlet distribution via implicit reparameterization (Figurnov et al., 2018).

### 4.3 TRAINING OBJECTIVE

We optimize a variational objective per token $x$:

$$\mathcal{L}_{\textbf{DirMoE}}(x) = \underbrace{- \mathbb{E}_{q_\phi(\tilde{\mathbf{z}}|x)}\big[\log p_\psi\big(x \mid \mathbf{r}(x)\big)\big]}_{\text{reconstruction on } \mathbf{r}(x)} + \beta_\theta \underbrace{\mathbb{E}_{q_\phi(\tilde{\mathbf{z}}|x)}\Big[D_{\mathrm{KL}}\big(\mathrm{Dir}(\boldsymbol{\alpha}^{(q)}(x, \tilde{\mathbf{z}})) \,\|\, \mathrm{Dir}(\boldsymbol{\alpha}^{(p)}(\tilde{\mathbf{z}}_{\mathrm{sg}})))\Big]}_{\text{Dirichlet KL}}$$
$$+ \underbrace{\mathcal{R}_{\text{sparsity}}(x)}_{\text{sparsity}} . \tag{8}$$

where $\beta_\theta \geq 0$ weights the slab KL, $\tilde{\mathbf{z}}_{\mathrm{sg}} = \mathrm{stopgrad}(\tilde{\mathbf{z}})$. We optimize a $\beta$-weighted variational objective (Higgins et al., 2017; Alemi et al., 2017) (Eq. (8)). When $\beta_\theta = 1$, the objective reduces to the standard ELBO. The decoder is a single network $g_\psi : \Delta^{E-1} \to \mathbb{R}^d$ applied to the latent $\mathbf{r}$. The details of the decoder and different loss terms are in Appendix A.2. So, we train the model with the following objective function:

$$\mathcal{L}_{\text{total}} = \mathcal{L}_{\text{LM}} + \mathcal{L}_{\textbf{DirMoE}} \tag{9}$$

**Sparsity regularization.** We encourage $k$ active experts in expectation per token using:

$$\mathcal{R}_{\text{sparsity}}(x) = \lambda_{\text{sparsity}} \Big( \sum_{i=1}^E \tilde{z}_i(x) - k \Big)^2, \tag{10}$$

where $\lambda_{\text{sparsity}} \geq 0$ controls sparsity strength. You can find the connection to the KL of the spike term in A.1. Figure 2 shows the importance of Regularization loss on the desired sparsity $(1 - k/E)$. Further experimental results on the importance of $R_{sparsity}$ in Appendix B.1.

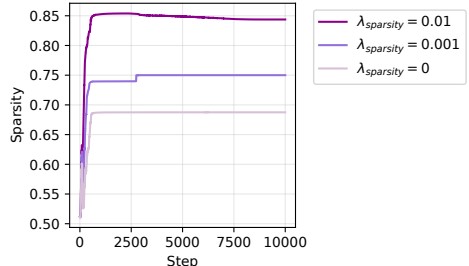

### 4.4 SCHEDULER

We discussed in Sec. 4.2 that $\boldsymbol{\theta} \sim \mathrm{Dir}(\boldsymbol{\alpha})$ with two concentration levels: $\alpha_{\mathrm{hi}}$ for active experts and $\alpha_{\mathrm{lo}}$ for inactive ones, scaled by $\lambda^{(p)} > 0$. Let $S = \{i : z_i = 1\}$ with $|S| = k$ denote the (idealized) active set.[1] Under a Dirichlet, the sum over any subset is Beta-distributed; therefore

Figure 2: Effect of sparsity regularization on the sparsity, there is less sparsity compared to the desired sparsity with lower $\lambda_{sparsity}$.

$$T := \sum_{i \in S} \theta_i \sim \mathrm{Beta}\big(k\lambda^{(p)}\alpha_{\mathrm{hi}}, (E-k)\lambda^{(p)}\alpha_{\mathrm{lo}}\big), \qquad \mathbb{E}[T] = \frac{k\alpha_{\mathrm{hi}}}{k\alpha_{\mathrm{hi}} + (E-k)\alpha_{\mathrm{lo}}} =: m. \tag{11}$$

Solving for the ratio $r := \alpha_{\mathrm{hi}}/\alpha_{\mathrm{lo}}$ yields

$$r = \frac{m}{1-m} \cdot \frac{E-k}{k}. \tag{12}$$

Hence, for a desired expected probability mass fraction $m$ on the $k$ active experts, one can fix $r$ by equation 12 and then schedule the absolute scales separately.[2]

**Temperature schedule.** We use a short schedule for the gate temperature to move from exploratory to decisive selection variables:

$$\tau_z^{(t)} = \max\big(\tau_{\min}, \tau_0 \rho^{\,t}\big). \tag{13}$$

Large $\tau_z$ early encourages exploration; small $\tau_z$ later yields near-binary $\tilde{\mathbf{z}}$.

---

[1] We use this discrete view to calibrate the prior; in practice we train with relaxed gates $\tilde{\mathbf{z}}$, cf. Sec. 4.2.

[2] Note that $\mathbb{E}[T]$ is independent of $\lambda^{(p)}$; changing $\lambda^{(p)}$ alters dispersion (and thus sparsity tendencies) but not the mean probability mass split between active and inactive sets when $r$ is fixed.

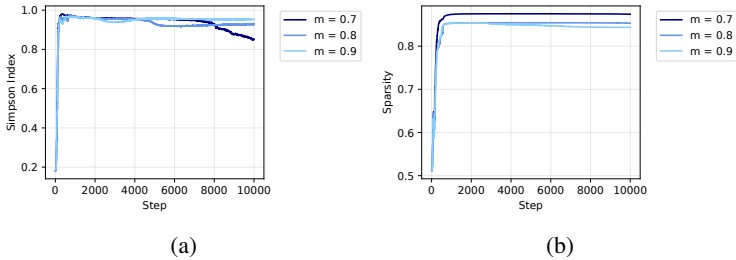

Figure 3: Effect of $m$ and $\lambda$ on (a) Simpson index and (b) sparsity.
$\lambda$ is calculated from $m$ based on equation 16

**Dirichlet-parameter schedule.** We choose $r = \alpha_{\text{hi}}/\alpha_{\text{lo}}$ from equation 12 and keep it fixed to preserve the target mean probability mass $m$ on the active set. We then reduce the spike floor and tighten overall concentration during training:

$$\alpha_{\text{lo}}^{(t)} = \max\big(\alpha_{\text{floor}}, \alpha_{\text{lo}}^{(0)} \gamma^t\big), \qquad \alpha_{\text{hi}}^{(t)} = r\,\alpha_{\text{lo}}^{(t)}, \qquad \lambda^{(p,t)} = \lambda^{(p,0)}\,\eta^t, \tag{14}$$

with decay factors $\gamma, \eta \in (0,1)$ and a small numerical floor $\alpha_{\text{floor}}$ for stability in digamma($\Gamma$) evaluations.

## 5 DISCUSSION

### 5.1 CONTROL OF SPARSITY

We control sparsity with two orthogonal knobs: the mask temperature $\tau_z$ (exploration $\rightarrow$ decisiveness) and the Dirichlet concentration scale $\lambda$ (dispersion on the active simplex face). Early exploration reduces rich-get-richer collapse without extra load-balancing losses; later, a small $\tau_z$ yields a near-binary expert selection vector and better specialization.

### 5.2 CALIBRATED SPARSITY VIA THE DIRICHLET CONCENTRATION

We introduce the Dirichlet concentration $\lambda$ as a **calibrated sparsity knob** for the routing distribution in our MoE model. We use the **Simpson index** $H(\mathbf{p}) = \sum_i p_i^2$ as our sparsity metric; a higher value of $H(\mathbf{p})$ indicates a more concentrated (and thus sparser) distribution. You can find the role of $\lambda$ on the variance of the Dirichlet distribution (expert contribution) in Appendix A.6.

### 5.3 THEORETICAL JUSTIFICATION

We formally justify using $\lambda$ as a sparsity knob through a key theorem. The core idea is that the expected sparsity, as measured by the Simpson index, is a monotonic function of the Dirichlet concentration. We justify using the Dirichlet concentration $\lambda$ as a calibrated sparsity knob via the Simpson concentration $H(\mathbf{p}) = \sum_{i=1}^{E} p_i^2$.

**Lemma 1** (Expected Simpson index under Dirichlet). *Let* $\mathbf{p} \sim \text{Dir}(\lambda\boldsymbol{\beta})$ *with* $\boldsymbol{\beta} \in \mathbb{R}_{>0}^E$, $B = \sum_{i=1}^{E} \beta_i$, *and* $S_2 = \sum_{i=1}^{E} \beta_i^2$. *Then*

$$\mathbb{E}[H(\mathbf{p})] = \frac{\lambda S_2/B + 1}{\lambda B + 1}.$$

Proof is in Appendix A.3.

**Theorem 1** (Monotone sparsity control by concentration). *Fix* $\boldsymbol{\beta} \in \mathbb{R}_{>0}^E$ *with* $E \geq 2$ *and let* $\mathbf{p} \sim \text{Dir}(\lambda\boldsymbol{\beta})$. *Then* $f(\lambda) := \mathbb{E}[H(\mathbf{p})]$ *is strictly decreasing in* $\lambda > 0$, *with limits*

$$\lim_{\lambda \to 0^+} f(\lambda) = 1 \qquad and \qquad \lim_{\lambda \to \infty} f(\lambda) = \sum_{i=1}^{E} m_i^2,$$

where $m_i = \beta_i/B$ is the mean routing distribution. Proof is in Appendix A.4.

**Corollary 1** (Symmetric base). *If $\beta_i \equiv 1$, then $m_i = 1/E$ and Theorem 1 gives $f(\lambda)$ strictly decreasing from 1 to $1/E$. The calibration in Lemma 1 reduces to $\mathbb{E}[H] = \dfrac{\lambda + 1}{\lambda E + 1}$, implying $\lambda = \dfrac{1 - h}{hE - 1}$ for a target $h \in (1/E, 1)$.*

## 5.4 PRACTICAL CALIBRATION SPARSITY KNOB

Based on this theoretical foundation, we provide two practical ways to set the Dirichlet concentration to achieve a desired level of sparsity.

**(A) Symmetric Base Calibrator.** For a symmetric base where all experts are treated equally ($\beta_i = 1$), the expression simplifies, allowing us to directly map a target Simpson index $h \in (1/E, 1)$ to the required concentration $\lambda$. This is useful for initial exploration where no prior knowledge about expert specialization is assumed.

$$\lambda = \frac{1 - h}{hE - 1} \tag{15}$$

**(B) Two-Group (Beta) Calibrator.** For our method, we can model the total probability mass on a group of $s$ experts with high shape parameters ($\alpha_{\text{hi}}$) and $E - s$ experts with low ones ($\alpha_{\text{lo}}$). The total probability mass on the 'high' group follows a Beta distribution. We can set $\lambda$ to target a specific variance, $v_{\text{tar}}$, for this mass, offering a more nuanced control.

$$\lambda = \frac{\frac{m(1-m)}{v_{\text{tar}}} - 1}{s\alpha_{\text{hi}} + (E - s)\alpha_{\text{lo}}} \tag{16}$$

This approach is highly useful when we have a fixed number of active experts in mind (e.g., $s = k$) and want to control how concentrated the probability mass is on that group. You can find the derivation for Equation equation 16 in Appendix A.5.

**Which $\lambda$ to tune?** Routing depends on the posterior draw, so sparsity of $\mathbf{r}$ is controlled by the posterior scale $\lambda_q$ (in $\boldsymbol{\alpha}^{(q)}$). The prior scale $\lambda^{(p,t)}$ shapes regularization via the KL. In practice, we fix $\lambda_q$ via the calibrator (to hit a target $h$) and mildly decay $\lambda^{(p,t)}$ to encourage decisiveness. We investigate the $\lambda$ and m that are calculated based on Equation equation 16 for variance 0.01. As $m$ decreases (less mean mass probabilities on the active set), the $\lambda$ from Equation equation 16 increases to keep the Beta distribution variance fixed; This preserves a similar level of sparsity (Fig. 3b) while adjusting expert contribution on the active experts (reflected by $H(\theta)$ in Fig. 3a).

## 6 IMPLEMENTATION

**Architecture** Our backbone is LLaMA (Touvron et al., 2023) with 185M parameters, configured with 12 layers, RMSNorm (Zhang & Sennrich, 2019), SwiGLU activations (Shazeer, 2020), and rotary position embeddings (RoPE) (Su et al., 2021) and grouped-query attention (GQA) (Ainslie et al., 2023) with 4 query groups. We replace the vanilla Top-$k$ router with **DirMoE**, keeping the same expert counts and per-expert dimensions. The training follows Algorithm 1. Detailed implementation information is in Appendix C.1.

---

**Algorithm 1** DirMoE algorithm

---

**Require:** Batch $\{x_j\}_{j=1}^N$, experts $\{E_i\}_{i=1}^E$, router logits head $\ell(x)$, posterior base heads $\hat{\alpha}_{\text{hi}}(x), \hat{\alpha}_{\text{lo}}(x)$

**Require:** Scales: posterior $\lambda^{(q)}$, prior $\lambda^{(p,t)}$; temperature $\tau_z^{(t)}$; target $k$

1: Update schedules: $\tau \leftarrow \tau_z^{(t)}, \ \lambda^{(p)} \leftarrow \lambda^{(p,t)}$              (Eqs. 13, 14)
2: **for** each token embedding $x$ **do**
3:     **Relaxed gates** $\tilde{\mathbf{z}}$ via Gumble-sigmoid             (Equation 3)
4:     **Posterior slab** $\boldsymbol{\alpha}^{(q)}(x, \tilde{\mathbf{z}})$                  (Equation 4)
5:     **Prior slab** $\boldsymbol{\alpha}^{(p)}(\tilde{\mathbf{z}}_{\text{sg}})$                    (Equation 5)
6:     Sample $\boldsymbol{\theta} \sim \text{Dir}(\boldsymbol{\alpha}^{(q)})$              (implicit reparam.)
7:     **Router** $\mathbf{r} = \text{normalize}(\tilde{\mathbf{z}} \odot \boldsymbol{\theta}) \in \Delta^{E-1}$        (Equation 6)
8:     MoE output $y = \sum_{i=1}^E r_i \, E_i(x)$             (Equation 7)
9: **end for**
10: **Objective:** optimize the loss in equation 8.

---

**Training settings**   We train the model on The Pile dataset (Gao et al., 2021). We use Byte-Pair Encoding (BPE) tokenization (Sennrich et al., 2016) and train on ∼30B tokens. The zero-shot experiments use 60k steps (covering all 30B tokens), and ablation experiments run for 10k steps. Our implementation builds on Megatron-LM (Shoeybi et al., 2019). Megatron's MoE stack supports expert parallelism with all-to-all token dispatch and grouped-GEMM expert compute for efficiency (NVIDIA, 2025; Hejazi, 2024; Rajbhandari et al., 2022; Gale et al., 2023). We use the AdamW optimizer (Loshchilov & Hutter, 2019) with a learning-rate schedule combining cosine annealing (Loshchilov & Hutter, 2017) and linear warmup (Goyal et al., 2017). All experiments use 4-8 NVIDIA H100 GPUs. Hyperparameters are listed in Appendix C.3.

## 7 Experiments

### 7.1 Scalability

In this section, we compare the scalability of the **DirMoE** with Vanilla MoE. In Table 1, we show the computation cost in terms of throughput and iteration time; **DirMoE** has matching or exceeding the vanilla MoE baseline under compute parity. Because both systems utilize all-to-all token dispatch and grouped-GEMM expert compute, the observed efficiency is primarily determined by the global tokens-per-step and sequence length; our method introduces no additional bottlenecks (less than 1 %). Figure 4 demonstrates that our method has reliable training and achieves the desired sparsity for different $k$ and the number of experts. You can find further results on scalability in Appendix B.4.

Table 1: Training efficiency on LLaMA-185M with $E$=8, $k$=1.

| Method | Iteration time (ms)↓ | Throughput (TFLOP/s/GPU)↑ |
|---|---|---|
| Vanilla MoE (Switch) | 431.5 | 138.2 |
| DirMoE (ours) | 437.3 | 137.2 |

### 7.2 Zero-shot accuracy

We report the result of zero-shot accuracy in Table 2. We evaluate our method in seven widely used benchmarks. AR-Challenge (ARC-C) and ARC-Easy (ARC-E): grade-school science multiple-choice with adversarial hard and easier subsets (Clark et al., 2018); BoolQ: natural yes/no questions from real search queries (Clark et al., 2019); HellaSwag: common sense sentence completion with adversarially filtered decoys (Zellers et al., 2019); LAMBADA broad context next-word prediction requiring long range discourse understanding (Paperno et al., 2016); PIQA: physical common sense about everyday actions (Bisk et al., 2020); and RACE: reading comprehension from English exams (Lai et al., 2017). **DirMoE** yields superior or similar results to other methods; on average, we outperform them.

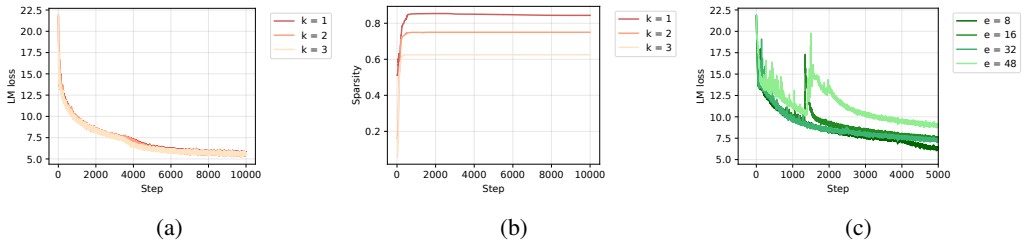

Figure 4: Effect of active experts ($k$) on (a) LLM loss and (b) sparsity; effect of number of experts (E) on (c) LLM loss.

Table 2: Zero-shot accuracy (%) on downstream tasks (higher is better). [†]denotes results reproduced by us.

| Method | ARC-c | ARC-e | BoolQ | HellaSwag | LAMBADA | PIQA | RACE | Avg. |
|---|---|---|---|---|---|---|---|---|
| Hash (Roller et al., 2021) | 19.28 | 45.45 | 54.95 | 29.68 | 31.44 | 63.06 | 27.66 | 38.79 |
| Lory (Zhong et al., 2024) | 20.31 | 42.97 | 49.54 | 28.75 | 32.35 | 62.24 | 27.75 | 37.70 |
| SparseMixer-v2 (Jin et al., 2023) | 19.80 | **46.72** | 45.96 | 30.24 | 34.12 | 62.89 | 29.00 | 38.39 |
| EC (Expert Choice) (Zhou et al., 2022) | 18.86 | 42.97 | 60.21 | 29.14 | 29.26 | 61.92 | 27.37 | 38.53 |
| Switch MoE[†] (Fedus et al., 2022) | 20.09 | 44.23 | 57.83 | 29.68 | 32.97 | 63.55 | 27.96 | 39.47 |
| ReMoE (Wang et al., 2025) | 20.22 | 46.68 | 54.16 | **30.26** | 35.94 | 63.55 | 29.38 | 40.03 |
| DirMoE (ours) | **20.57** | 46.20 | **61.52** | 29.93 | **36.44** | **63.75** | **29.52** | **41.13** |

## 7.3 EXPERT SPECIALIZATION

**DirMoE** controls routing sparsity without an explicit load-balancing loss by using the sparsity knob ($\lambda$) together with the near-binary expert selection vector. Although this probabilistic allocation can yield modest load asymmetries, it encourages experts to concentrate on distinct token subpopulations. As illustrated in Figure 5, our method has better expert specialization across different layers compared to Vanilla MoE. The latter explicitly equalizes contributions, which stabilizes utilization but also homogenizes experts and weakens their semantic focus.

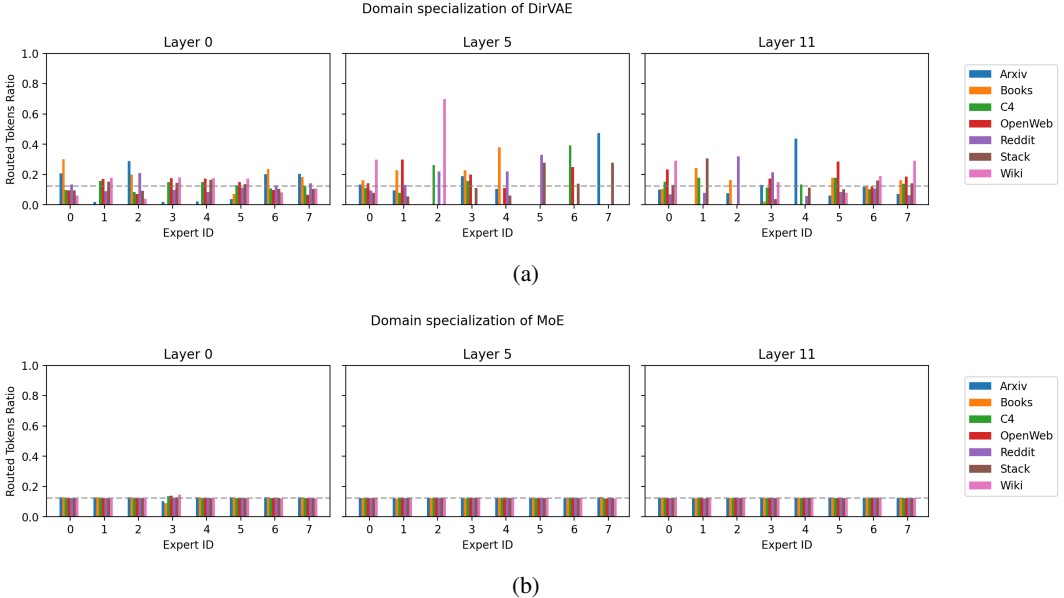

Figure 5: Domain specialization of (a) **DirMoE** and (b) Vanilla MoE in different layers and domains based on average routed token. The grey dashed line indicates the uniform distribution.

## 8 CONCLUSION AND FUTURE WORK

In this paper, we introduced **DirMoE**, a fully differentiable probabilistic router for Mixture-of-Experts that disentangles routing into (i) per-token expert selection and (ii) per-token probability allocation over experts (expert contribution). This white-box design gives explicit control over sparsity, enables calibrated concentration via Dirichlet parameters, and maintains end-to-end differentiability. Empirically, **DirMoE** achieves strong few-shot performance, and we ablate its components to study calibration and sparsity control. In future works, we envision two promising directions. Robustness: The probabilistic formulation and explicit concentration control may improve model robustness. Interpretability: the disentangled routing variables naturally expose which experts are active and how probability mass is allocated. This could yield clearer, more monosemantic expert specializations.

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

# A    ADDITIONAL DETAILS FOR METHOD AND PROOF

## A.1    CONNECTION OF $\mathcal{R}_{\text{SPARSITY}}$ TO TO THE SPIKE KL.

Let the variational spike be a mean-field Bernoulli $q(\mathbf{z} \mid x) = \prod_{i=1}^{E} \text{Bern}(z_i \mid p_i(x))$ with $p_i(x) = \sigma(\ell_i(x))$, and consider the spike prior $p_\pi(\mathbf{z}) = \prod_{i=1}^{E} \text{Bern}(z_i \mid \pi)$ with $\pi = \frac{k}{E}$ (or the fixed-size microcanonical prior uniform on $\{\mathbf{z} : \sum_i z_i = k\};$). The exact spike KL is

$$D_{\text{KL}}\big(q(\mathbf{z} \mid x) \,\big\|\, p_\pi(\mathbf{z})\big) = \sum_{i=1}^{E} \left[ p_i \log \tfrac{p_i}{\pi} + (1 - p_i) \log \tfrac{1-p_i}{1-\pi} \right] = \sum_{i=1}^{E} D_{\text{Ber}}(p_i \| \pi).$$

A second–order expansion of the Bernoulli KL around $p_i = \pi$ gives

$$D_{\text{Ber}}(p_i \| \pi) = \frac{(p_i - \pi)^2}{2\,\pi(1 - \pi)} + \mathcal{O}\big(|p_i - \pi|^3\big).$$

Therefore, locally,

$$D_{\text{KL}}(q \| p_\pi) \approx \frac{1}{2\,\pi(1 - \pi)} \sum_{i=1}^{E} (p_i - \pi)^2.$$

By Cauchy–Schwarz, $\sum_i (p_i - \pi)^2 \geq \frac{1}{E}\big(\sum_i (p_i - \pi)\big)^2 = \frac{1}{E}\big(\sum_i p_i - k\big)^2$, so we obtain the *local lower bound*

$$D_{\text{KL}}(q \| p_\pi) \gtrsim \frac{\big(\sum_{i=1}^{E} p_i(x) - k\big)^2}{2E\,\pi(1 - \pi)}. \tag{17}$$

Hence the token–local sparsityinality penalty is a smooth surrogate for the spike KL:

$$\mathcal{R}_{\text{Sparsity}}(x) = \lambda_{\text{sparsity}} \left( \sum_{i=1}^{E} \tilde{z}_i(x) - k \right)^2 \quad \text{with} \quad \lambda_{\text{sparsity}} \approx \frac{\beta_z}{2\,E\,\pi(1-\pi)}$$

matches the local curvature of $\beta_z \, D_{\text{KL}}(q \| p_\pi)$ when we replace the (intractable in backprop) $\sum_i p_i(x)$ by its low-variance stochastic estimator $\sum_i \tilde{z}_i(x)$.

**Remarks.** Using $\tilde{z}_i$ in $\mathcal{R}_{\text{sparsity}}$ gives $\mathbb{E}\big[(\sum_i \tilde{z}_i - k)^2\big] = \text{Var}\big(\sum_i \tilde{z}_i\big) + \big(\sum_i p_i - k\big)^2$, so the penalty also discourages high-variance gates; as $\tau_z \downarrow 0$, this gap shrinks. If desired, a deterministic variant $\mathcal{R}_{\text{Sparsity}}(x) = \lambda_{\text{sparsity}}\big(\sum_i p_i(x) - k\big)^2$ targets only the bias term.

## A.2    OBJECTIVE FUNCTION.

The decoder is a single network $g_\psi : \Delta^{E-1} \to \mathbb{R}^d$ applied to the latent $\mathbf{r}$:

$$\hat{x}(x) = g_\psi\big(\mathbf{r}(x)\big), \qquad p_\psi\big(x \mid g_\psi(\mathbf{r})\big) = \mathcal{N}\big(g_\psi(\mathbf{r}), \sigma^2 I\big),$$

so, the reconstruction term reduces to scaled MSE

$$-\log p_\psi\big(x \mid g_\psi(\mathbf{r}(x))\big) = \frac{1}{2\sigma^2} \big\| x - g_\psi(\mathbf{r}(x)) \big\|_2^2 + \frac{d}{2} \log(2\pi\sigma^2),$$

and we drop the constant and often set $\sigma^2 = 1$ in practice. With the posterior $q_\phi(\boldsymbol{\theta} \mid x, \tilde{\mathbf{z}}) = \text{Dir}\big(\boldsymbol{\alpha}^{(q)}(x, \tilde{\mathbf{z}})\big)$ and the slab prior $p(\boldsymbol{\theta} \mid \tilde{\mathbf{z}}) = \text{Dir}\big(\boldsymbol{\alpha}^{(p)}(\tilde{\mathbf{z}}_{\text{sg}})\big)$, the KL term inside equation 8 is

$$D_{\text{KL}}\Big(\text{Dir}\big(\boldsymbol{\alpha}^{(q)}(x, \tilde{\mathbf{z}})\big) \,\Big\|\, \text{Dir}\big(\boldsymbol{\alpha}^{(p)}\big)\Big) = \log\Gamma\Big(\sum_i \alpha_i^{(q)}\Big) - \sum_i \log\Gamma(\alpha_i^{(q)}) - \log\Gamma\Big(\sum_i \alpha_i^{(p)}\Big)$$

$$+ \sum_i \log\Gamma(\alpha_i^{(p)}) + \sum_{i=1}^{E} \big(\alpha_i^{(q)} - \alpha_i^{(p)}\big)\Big(\psi(\alpha_i^{(q)}) - \psi\Big(\sum_j \alpha_j^{(q)}\Big)\Big), \tag{18}$$

with $\psi$ the digamma function and where

$$\alpha_i^{(q)}(x, \tilde{\mathbf{z}}) = \lambda^{(q)}\Big(\tilde{z}_i\,\alpha_{\text{hi},i}(x) + (1 - \tilde{z}_i)\,\alpha_{\text{lo},i}(x)\Big), \qquad \alpha_i^{(p)}(\tilde{\mathbf{z}}_{\text{sg}}) = \lambda^{(p,t)}\Big(\tilde{z}_i^{\text{sg}}\,\alpha_{\text{hi}} + (1 - \tilde{z}_i^{\text{sg}})\,\alpha_{\text{lo}}\Big).$$

## A.3 EXPECTED SIMPSON INDEX UNDER DIRICHLET PROOF

Here, we provide proof for Lemma 1:

*Proof sketch.* For a Dirichlet with parameters $\boldsymbol{\alpha}$, $\mathbb{E}[p_i^2] = \frac{\alpha_i(\alpha_i+1)}{\alpha_0(\alpha_0+1)}$ with $\alpha_0 = \sum_i \alpha_i$. Summing yields:

$\mathbb{E}[H] = \frac{\sum_i (\alpha_i^2 + \alpha_i)}{\alpha_0(\alpha_0+1)}$. Setting $\alpha_i = \lambda\beta_i$ gives $\mathbb{E}[H] = \frac{\lambda^2 S_2 + \lambda B}{(\lambda B)(\lambda B+1)} = \frac{\lambda S_2 + B}{B(\lambda B+1)} = \frac{\lambda S_2/B+1}{\lambda B+1}$. $\qquad\square$

## A.4 MONOTONE SPARSITY CONTROL BY CONCENTRATION PROOF

Here, we provide proof for Lemma 1:

*Proof.* By Lemma 1, $f(\lambda) = \dfrac{a\lambda + 1}{b\lambda + 1}$ with $a = S_2/B$ and $b = B$. Differentiating gives $f'(\lambda) = \dfrac{a - b}{(b\lambda + 1)^2}$. Since $\boldsymbol{\beta} > 0$ and $E \geq 2$, we have $B^2 = \sum_i \beta_i^2 + 2\sum_{i<j} \beta_i\beta_j > \sum_i \beta_i^2 = S_2$, hence $a - b = \frac{S_2 - B^2}{B} < 0$, so $f'(\lambda) < 0$ for all $\lambda > 0$. The limits follow directly from the rational form: $f(0^+) = (0+1)/(0+1) = 1$ and $f(\infty) = a/b = (S_2/B)/(B) = S_2/B^2 = \sum_i m_i^2$. $\qquad\square$

## A.5 TWO-GROUP BETA CALIBRATOR DERIVATION

Here, we provide details on how to derive Equation equation 16. Let $S$ be the active set with $|S| = s$ (for our setup $s=k$). With the two-level Dirichlet $\boldsymbol{\theta} \sim \mathrm{Dir}(\lambda\boldsymbol{\beta})$, where $\beta_i = \alpha_{\mathrm{hi}}$ for $i \in S$ and $\beta_i = \alpha_{\mathrm{lo}}$ otherwise, the total mass on the active group

$$T := \sum_{i \in S} \theta_i$$

is Beta-distributed by equation 11:

$$T \sim \mathrm{Beta}(a, b), \qquad a = \lambda s \alpha_{\mathrm{hi}}, \quad b = \lambda (E - s) \alpha_{\mathrm{lo}}.$$

Hence

$$\mathbb{E}[T] = \frac{a}{a+b} = \frac{s \alpha_{\mathrm{hi}}}{s \alpha_{\mathrm{hi}} + (E-s) \alpha_{\mathrm{lo}}} =: m,$$

and using $\mathrm{Var}(\mathrm{Beta}(a,b)) = \dfrac{ab}{(a+b)^2(a+b+1)}$,

$$\mathrm{Var}(T) = \frac{m(1-m)}{a+b+1} = \frac{m(1-m)}{\lambda C + 1}, \qquad C := s \alpha_{\mathrm{hi}} + (E-s) \alpha_{\mathrm{lo}}.$$

Solving for $\lambda$ at a target variance $v_{\mathrm{tar}} \in (0, m(1-m))$ yields

$$\lambda = \frac{\dfrac{m(1-m)}{v_{\mathrm{tar}}} - 1}{s \alpha_{\mathrm{hi}} + (E-s) \alpha_{\mathrm{lo}}}. \tag{19}$$

## A.6 VARIANCE AND THE ROLE OF $\lambda$.

Let $\alpha_0 = \sum_i \alpha_i^{(p)}$. For a Dirichlet,

$$\mathrm{Var}(\theta_i) = \frac{\alpha_i^{(p)}(\alpha_0 - \alpha_i^{(p)})}{\alpha_0^2(\alpha_0 + 1)}.$$

Scaling all concentrations by $\lambda$ scales $\alpha_0$ by $\lambda$, leaves means unchanged, and shrinks variances roughly as $1/(\lambda \alpha_0)$. Hence, decreasing $\lambda$ pushes draws toward vertices (sparser allocations) without altering the mean split $m$ induced by the base ratios.

# B  EXPERIMENTS RESULTS

## B.1  EFFECT OF $\lambda_{sparsity}$

We further investigate the effect of $R_{sparsity}$ on LLM loss, Simpson index and $\tilde{z}$ in Figure 6

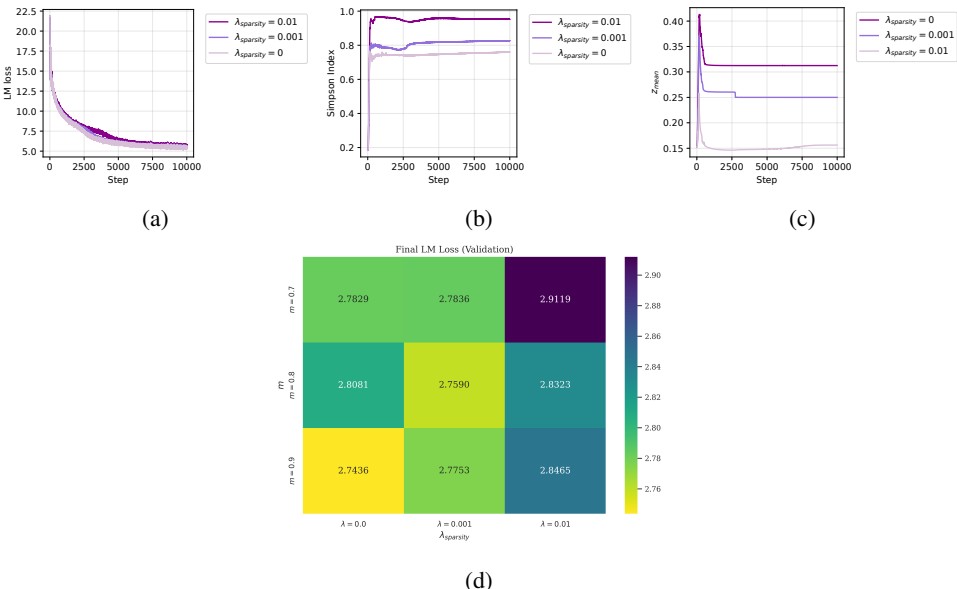

Figure 6: Effect of $\lambda_{sparsity}$ on (a) LLM loss and (b) Simpson index, and (c) $\tilde{z}$ (d) heatmap of Loss.

We further investigate the effect of $m$ and $\lambda_{sparsity}$ on LLM loss in Fig. 6d). Intuitively, increasing $m$ concentrates more probability mass on the selected experts. If the sparsity weight $\lambda_{\text{sparsity}}$ is set to zero, the router can activate more than $k$ experts on difficult tokens. This often lowers the LM loss (more capacity per token) but reduces efficiency because the effective sparsity decreases. In practice, we use a small $\lambda_{\text{sparsity}} > 0$ so the model benefits from early accuracy gains while converging back toward the desired $k$ for efficiency.

## B.2  STABILITY ANALYSIS OF ACTIVE EXPERTS

Here, we demonstrate the average number of active experts, the maximum number of active experts, and the gradient norm throughout our training and in Figure 7

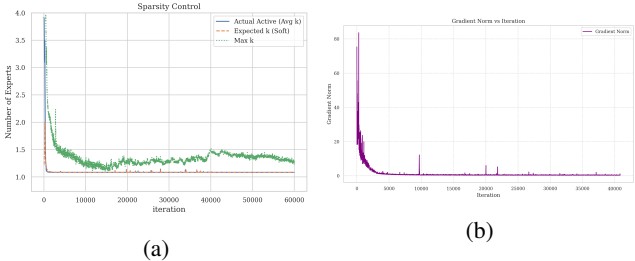

Figure 7: (a) average and maximum number of active experts (b) gradient norm.

## B.3  ABLATION OF VAE LOSS

Here, we further investigate the importance of the reconstruction loss and VAE loss on LLM loss, $\alpha^q$ and $\tilde{z}$ in Figure 8

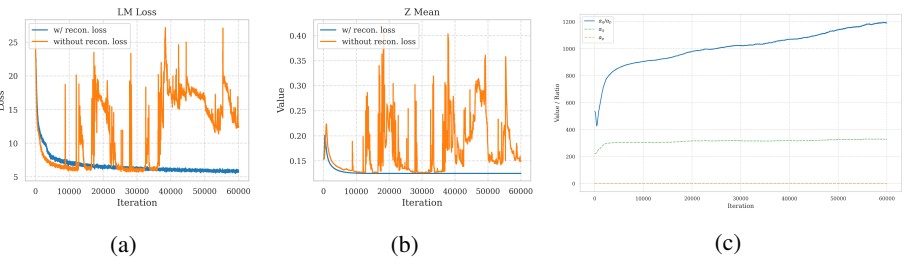

(a)          (b)          (c)

Figure 8: Effect of reconstruction loss on (a) LLM loss and (b) $\tilde{z}$, and (c) KL loss effect on $\alpha^q$.

'

### B.4 SCALABILITY RESULTS

We explored the effect of desired active experts on $\tilde{z}$ and Simpson index in Figure 9.

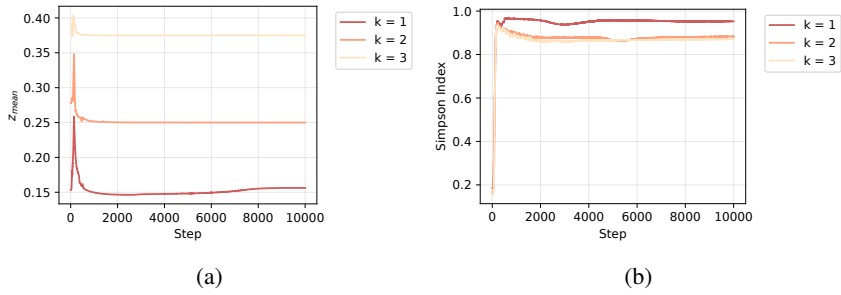

(a)          (b)

Figure 9: Effect of active experts ($k$) on (a) $\tilde{z}$ and (b) Simpson index.

## C DETAILS ON TRAINING

### C.1 IMPLEMENTATION DETAILS

**Capacity and kernel considerations.** Near-binary gates and sharp allocations increase per-step variance in expert loads; if the stack drops overflow tokens (capacity-based routing), this can introduce training noise and dead-expert effects. Two practical mitigations are standard: dropless and block-sparse kernels (e.g., MegaBlocks), which eliminate token drops by packing variable expert batches into high-utilization grouped GEMMs (Gale et al., 2023), or using a sufficiently high capacity factor (CF) in standard MoE stacks to reduce drops at the cost of extra padding or compute (Rajbhandari et al., 2022). Both approaches stabilize optimization while preserving the benefits of sparsity control from $\tau_z$ and $\lambda$. We used the first approach.

### C.2 PRACTICAL IMPLEMENTATION NOTES

**Systems.** Our design keeps training fully differentiable and avoids balancing losses; at scale, deploy with dropless kernels (Gale et al., 2023) or adequate capacity in expert-parallel engines (Rajbhandari et al., 2022).

### C.3 HYPERPARAMETERS

You can find Hyperparameters for training in Table 3 and hyperparameters for our router in Table 4.

Table 3: **Optimization Hyperparameters**

| Component | Parameter | Default Value |
|---|---|---|
| **Optimizer** | Optimizer | AdamW |
| | Learning rate | $1 \times 10^{-4}$ |
| | Weight decay | $1 \times 10^{-2}$ |
| | Adam betas / eps | (0.9, 0.995) / $1 \times 10^{-8}$ |
| | LR schedule | Warmup 1% steps $\rightarrow$ Cosine (min lr $= 0.1\times$ base) |
| | Global grad clip | 1.0 |
| | Precision | bf16 (router logits/Dirichlet in fp32) |
| **Batch** | Micro-batch size | 32 |
| | Grad accumulation | 512 |
| | Seed | 1234 |

Table 4: **Routing Hyperparameters**

| Component | Parameter | Default Value |
|---|---|---|
| **Gates (spike)** | Experts $E$ / target actives $k$ | $E{=}8,\ k{=}1$ |
| | Gate temperature $\tau_z$ schedule equation 13 | $2.0 \rightarrow 0.3$ (cosine) |
| | Gate bias init (per logit) | $\tau_0 \cdot \mathrm{logit}(k/E)$ |
| | Per-token centering | On ($\ell \leftarrow \ell - \mathrm{mean}_i(\ell)$) |
| | $z$-threshold for routing map (metrics) | 0.125 |
| **Slab (Dirichlet)** | Posterior scale $\lambda^{(q)}$ (Eq. (16)) | $\lambda^{(q)} = 20$ |
| | Prior mean mass $m$ on actives equation 11 | $m{=}0.9$ |
| | Ratio $r{=}\alpha_{\mathrm{hi}}/\alpha_{\mathrm{lo}}$ equation 12 | $r\approx39.7$ (for $E{=}8, k{=}1, m{=}0.85$) |
| | Prior $\alpha_{\mathrm{lo}}^{(0)}$ / floor equation 14 | 0.005 |
| | Prior $\alpha_{\mathrm{hi}}^{(0)}{=}r\,\alpha_{\mathrm{lo}}^{(0)}$ | $\approx 1.99$ |
| | Prior scale $\lambda^{(p)}$ schedule equation 14 | $0.5 \rightarrow 0.3$ (slow exp/cosine) |
| | KL weight $\beta_\theta$ | $1 \times 10^{-2}$ |
| | Recon variance $\sigma^2$ (Sec. 4.3) | 1.0 |
| **Router output** | Leak before normalize | $\varepsilon_w{=}1 \times 10^{-3}$ |
| | Epsilon clamp | Off |
| | Expected-$k$ penalty equation 10 | $\lambda_{\mathrm{sparsity}} : 0.01$ |

# D  AI USAGE CLARIFICATION

We used AI-assisted tools solely for editing and syntax lookups. All ideas, methods, experiments, analyses, and conclusions were conceived and written by the authors. Every AI suggestion was manually reviewed and edited before applying. No text, figures, tables, code, or data were generated or analyzed autonomously by AI. The authors retain full responsibility for the content.

