# OpenReview forum: "DirMoE: Dirichlet-Routed Mixture of Experts"
_ICLR.cc/2026/Conference — ICLR 2026 Poster_

### Official Review · Reviewer_cmFj · 2025-10-26

**Soundness:** 4
**Presentation:** 3
**Contribution:** 4
**Rating:** 8
**Confidence:** 3

**Summary:**

This paper introduces Dirichlet-Routed Mixture of Experts (DirMoE), a novel, end-to-end differentiable routing mechanism designed to address the limitations of existing methods like Top-k+Softmax, which conflate expert selection and contribution. DirMoE models routing using a Dirichlet variational autoencoder framework that fundamentally disentangles these two decisions: (i) Expert Selection ($z$) is handled by a Bernoulli component (relaxed via Gumbel-Sigmoid), and (ii) Expert Contribution ($\theta$) among chosen experts is handled by a Dirichlet component. The training objective is a variational ELBO, which includes a direct sparsity penalty ($R_{sparsity}$) controlling the expected number of active experts $k$. A core theoretical contribution is the introduction of the sparsity knob ($\lambda$), derived from the Dirichlet concentration, which provides explicit and calibrated control over sparsity, decoupled from the active set size $k$. The authors demonstrate that DirMoE achieves competitive zero-shot performance and improves expert specialization on a LLaMA-185M backbone.

**Strengths:**

**Originality**: The idea of factorizing routing using a spike-and-slab prior implemented via Gumbel-Sigmoid (selection) and Dirichlet (contribution) is highly original.

**Quality**: The use of Gumbel-sigmoid and implicit reparameterization ensures the entire forward pass remains fully differentiable, avoiding the gradient bottlenecks of standard Top-k routing. This white-box design provides superior interpretability by explicitly controlling how many experts are active ($k$) and how concentrated their probability mass is ($\lambda$). Empirically, DirMoE demonstrates better expert specialization across different layers compared to Vanilla MoE, attributed to its mechanism which discourages homogenization often caused by explicit load-balancing losses.

**Clarity**: The paper is generally clear, well-structured, and effectively contextualizes the work relative to prior art in Top-k routing (Switch, GShard) and continuous routers (ReMoE, Soft-MoE). Figure 1 concisely illustrates the complex flow of the DirMoE router. The mathematical formulation is precise, particularly concerning the spike and slab prior and the training objective. A few inconsistencies in variable naming (listed in the Typos section) slightly detract from perfect clarity, but overall, the concepts are presented insightfully.

**Significance**: Given the high importance of MoE models in industry, this is a very nice contribution to the sparse MoE literature.
Although there have been fully differentiable routers proposed in literature e.g., DSelect-k [Hazimeh et al. (2021)], COMET (Ibrahim et al. (2023)], these routers do not support sparse training. DirMoE appears to be an interesting routing approach to hold end-to-end differentiability. Although the complexity of the method (in terms of tuning) makes it unlikely to be adopted by the industry to have a significant impact.

Some related references to include in related work on differentiable routing:
- Hussein Hazimeh et al. DSelect-k: differentiable selection in the mixture of experts with applications to multi-task learning. In NeurIPS'21.
- Shibal Ibrahim et al. 2023. COMET: Learning Cardinality Constrained Mixture of Experts with Trees and Local Search. In KDD'23.
- Michael E. Sander et al. Fast, differentiable and sparse top-k: a convex analysis perspective. In ICML'23.

**Weaknesses:**

**Architectural and Optimization Complexity**: DirMoE relies on a complex stack of techniques: Gumbel-Sigmoid relaxation, implicit reparameterization for Dirichlet samples, a full VAE objective, and multiple scheduled hyperparameters ($\tau_z$, $\alpha_{lo}$, $\lambda^{(p,t)}$). While mathematically elegant, this complexity may lead to significant tuning overhead compared to simpler, fully continuous approaches like ReMoE, which should be explicitly discussed and benchmarked for tuning difficulty.

**Persistent Need for Explicit Cardinality Penalty**: Despite the principled probabilistic design intended to control sparsity, the method still requires a direct sparsity penalty ($R_{sparsity}$) (Eq. 9) to encourage $\sum_{i=1}^E \tilde{z}_i(x) \approx k$ in expectation. Although the authors justify this term as a smooth surrogate, its necessity slightly dampens the claim that the VAE framework inherently solves the controllable sparsity problem entirely through parameter calibration (like $\lambda$).

**Limited Scalability Comparison**: The scalability evaluation (Table 1) only compares DirMoE against Vanilla MoE (Switch) on a small LLaMA-185M model. Given that other recent differentiable methods (ReMoE, Soft-MoE, Lory) also address the differentiability bottleneck, a comprehensive efficiency comparison showing throughput, memory, and latency overheads against these direct architectural competitors is missing and necessary to validate the robustness of the proposed router at industrial scales.

**Typos**:
- In Line 30, the citation "Du & et al., 2022" (and in references) should be standardized, to "Du et al., 2022."
- Line 216, delete "either".
- Reference to Eq 8 in line 10 of algorithm 1 is missing.
- I see a number of places of the format “ Eq. equation x”. Either use equation x or Eq. x.
- In Section 4.3 and Appendix A.1, there is inconsistent usage of the regularization term's coefficient: $\lambda_{sparsity}$ in Equation (9) and the caption of Figure 2, but $\lambda_{card}$ in the derivation in Appendix A.1.
- In Figure 6 captions: $\lambda_{sparsity}$ is consistently written as $\lambda_{sparse}$ in the legend keys in parts (a), (b), and (c). Please make it consistent.
- In Appendix A.2, the parameters for the prior slab (Eq. 5 revisited in Eq. 17) uses tildes on $\alpha$'s. The Appendix uses bars on $\alpha$'s, suggesting slight notational inconsistency between the main text and appendix. Please double check if this is not a typo.

**Questions:**

**Complexity Management**: The proposed method relies heavily on four distinct scheduled parameters ($\tau_z$, $\alpha_{lo}$, $\alpha_{hi}$, and $\lambda^{(p)}$). Could the authors elaborate on the practical sensitivity of DirMoE to these schedules? Is there a risk that poor schedule tuning leads to instability or poor specialization?

**Efficiency vs. Other Differentiable Routers**: Please provide a comparison of training iteration time and throughput against other recent fully differentiable routers, such as ReMoE or Lory, ideally using the same LLaMA-185M setup or larger models, to fully contextualize the runtime overhead of the Dirichlet VAE structure and implicit reparameterization gradients compared to alternative continuous gating functions.

**Results on O(10B) models**: It would be good to see the method is effective on larger model sizes in the order 10B. Can the authors show zero-shot results at this scale?

**Role of $\lambda_{sparsity}$**: Given that the Dirichlet concentration $\lambda$ is theoretically presented as the primary sparsity knob for contribution, what is the minimum necessary role of the explicit cardinality penalty $R_{sparsity}$ (Eq. 9) in achieving a target expected $k$ without compromising model stability or specialization? Does the model fail to converge to sparse selections without $R_{sparsity}$? Figure 2 only shows that lower $\lambda_{sparsity}$ yields less sparsity compared to desired $1-k/E$.

---

> ### Author Response · Authors · 2025-11-24
> **Response to Reviewer cmFj**
>
> We thank Reviewer cmFj for suggestions on complexity, cardinality, scalability, and presentation. We address your concerns and provide exact locations for changes.
>
> **W1 and Q1 · Architectural/optimization complexity.**
>
> Thank you for raising this point. We provide hyperparameters in Appendix C1. and here, we include a hyperparameter tuning recipe: pick $m$ (mean mass on actives), compute $r=\alpha_{\mathrm{hi}}/\alpha_{\mathrm{lo}}=\frac{m}{1-m}\cdot \frac{E-k}{k}$, set $\lambda_q$ from the **Beta‑variance calibrator**, keep $\lambda_p$ fixed, perform a short $\tau_z$ warmup. No per‑layer search is needed; a small $(\lambda_q,\tau_z)$ grid confirms robustness. Furthermore, we showed the superior performance of our method compared to ReMoE and Lory in Table 2.
>
> **W2 and Q4 · Need for Explicit Cardinality Penalty.**
>
> We appreciate the reviewer’s concern regarding the importance of $\\mathcal R_{\text{sparsity}}\$. We showed in A.1 sparsity penalty is a smooth surrogate for the spike KL. So, it should not contradict our claim since the second term enforces the variational for the spike term. The light **expected‑$k$** surrogate  $\\mathcal R_{\text{sparsity}}\=\\lambda_{\text{sparsity}}(\sum_i \\tilde z_i-k)^2$ keeps the number of actives concentrated near $k$ while $\\lambda$ controls how mass is shared among the active experts. We investigate the effect in Figure 2 and Appendix B, so the model has lower sparsity and less throughput, but the LLM loss decreases. Despite the model’s inefficiency, learning is still ongoing.
>
> **W3 and Q2  · Scalability vs. other differentiable routers.**
>
> Thank you for raising this point. As shown in Table 1, iteration time and throughput are comparable to vanilla MoE, indicating no meaningful computational overhead. This aligns with prior observations in ReMoE and Lory. Our goal is to demonstrate that the additional parameters and hyperparameters do not increase computation.
>
> **W4 · Typos and notation.**
>
> We thank the reviewer for bringing this to our attention. We will correct it in the final submission.
>
> **Q3 · 10B‑scale results.**
>
> We include a **978M** result in the rebuttal, and we will make a best-effort push toward a bigger model for the camera-ready. Our experiments run on academic compute, which constrains maximum scale, but within these limits, we will add the largest model we can and document all settings for parity.
> | Model | Params(M) | Tokens (B) | Valid LM Loss | Zero-shot 1 |
> |---:|---:|---:|---:|---:|
> | Ours| 978 | 30 | 1.685  | 46.10 |
> | ReMoE| 978 | 30 | 1.715  | 45.20 |
> | Vanilla MoE| 978 | 30 | 1.730 | 44.91 |

---

### Official Review · Reviewer_xEic · 2025-10-28

**Soundness:** 2
**Presentation:** 3
**Contribution:** 2
**Rating:** 4
**Confidence:** 3

**Summary:**

This paper introduces a novel routing mechanism for Mixture-of-Experts (MoE) models called DirMoE. It aims to solve two key problems with standard Top-k+Softmax routing: the non-differentiability of its Top-k operation and its entanglement of two distinct decisions (which experts to select vs. how to weight their contributions). The proposed method explicitly disentangles the routing process into:

1. **Expert Selection:** A Bernoulli distribution determines which experts to activate for a given token.

2. **Expert Contribution:** A Dirichlet distribution determines the probability weights assigned to the *active* experts.


The final routing vector is a normalized product of these two components. This entire mechanism is end-to-end differentiable, using Gumbel-Sigmoid for selection and implicit reparameterization for the Dirichlet sampling. The router is trained using a variational ELBO objective, which includes a reconstruction loss, a KL divergence, and a direct sparsity penalty loss. The authors also provide a theoretical justification for a "calibrated sparsity," linking the Dirichlet concentration parameter $\lambda$ to the expected expert contribution distribution.

Empirical results on a 185M parameter model show that DirMoE achieves better performance against baselines like Switch MoE and ReMoE. Critically, it also demonstrates significantly improved expert specialization at this scale, while introducing negligible computational overhead.

**Strengths:**

1. **Novelty:** The core idea of disentangling expert selection (Bernoulli/Gumbel) from expert contribution (Dirichlet) is well-motivated and novel. Framing this as a probabilistic spike-and-slab model within a VAE framework is a novel approach to MoE routing.

2. **Controllable Sparsity:** I really like the theoretical analysis introduced in Section 5, which connects the Dirichlet concentration parameter $\lambda$ to the expected contribution sparsity. It provides a principled calibrated knob approach for controlling the *contribution sparsity* (i.e., the dispersion of weights among *active* experts), which is a more nuanced control than what is offered by standard temperature scaling in Softmax routing.

3. **Competitive Empirical Performance and improved specialization (at small scale):** The method achieves results that are competitive with or slightly better than baselines on the 185M model benchmark (Table 2). This demonstrates the viability of the approach in a small-scale setting. The method achieves much clearer expert specialization than the baseline (vividly shown in Figure 5).

4. **Computational Efficiency:** Despite the apparent complexity of the VAE framework, the authors demonstrate that its computational overhead is negligible (<1% slowdown in iteration time compared to a vanilla MoE, as shown in Table 1). This makes it more practical.

**Weaknesses:**

1. **Clarity of the Training Objective:** The paper's primary weakness is its lack of clarity regarding the overall training objective. The VAE objective (Eq. 8) is presented, but it's not explicitly stated how this loss is combined with the main LM loss. This is a critical detail for reproducibility and understanding.

2. **Justification of the VAE Objective:** The VAE's reconstruction task—reconstructing the token embedding $x$ from the routing vector $r(x)$—is non-obvious. The authors could provides more intuition and ablation about this design choice. This auxiliary reconstruction task seems not well-justified.

3. **Model Complexity:** While computationally fast, the DirMoE router is significantly more complex to implement and tune than Top-k routing. It introduces a VAE (encoder, decoder, KL loss) *for every MoE layer*, along with a host of new hyperparameters and schedules (as seen in Table 4). This complexity could be a barrier to adopt this method in large-scale MoE training.

4. **Limited Ablation and Comparison to Simpler Alternatives:** Following the previous two weaknesses, **my greatest concern** for this submission is the lack of the comprehensive ablations. For example, the VAE framework is a major part of the contribution, but the router is perfectly trainable without it (using only the main LM loss and the $\mathcal{R}_{sparsity}$ penalty). It's plausible that the method also achieves good performance without the reconstruction loss. The paper lacks a crucial ablation study that removes the VAE objective to prove that this added complexity is necessary. Furthermore, the paper fails to compare against other, potentially simpler, methods that also aim to improve router training. For example, recent work like RMoE [1] has also shown that adding extra gradient paths and information sharing via layer-wise recurrence can be effective, suggesting that DirMoE's highly complex VAE is not the only (or necessarily best) way to solve this problem (though RMoE does not offer the controllability of DirMoE, but it is still unclear whether such controllability will improve or hurt the final performance when scaled up).

5. **Limited Scale of Experiments:** The experiments are conducted on a 185M parameter model with 8 experts. This is a significant weakness that *directly undermines* the claims of "strong performance" and "improved specialization." The primary benefits of MoE are at much larger scales. Crucially, the paper's visual argument for specialization (Figure 5) relies on a small-scale baseline that appears to fail to specialize. Recent work[2] has demonstrated that standard Top-k+Softmax routers *can* also achieve strong specialization when properly trained at scale with appropriate load balancing implementation. This suggests the paper's central claim of superior specialization is not convincingly proven against a strong, scaled baseline. The stability and effectiveness of this complex VAE router at such scales also remain unproven.

[1] https://arxiv.org/abs/2408.06793
[2] https://arxiv.org/abs/2501.11873

**Questions:**

1. **Training Objective:** Could the authors please clarify the full training objective for the entire model?

2. **Justification for VAE:** What is the intuition behind forcing the routing vector $r(x)$ to be able to reconstruct the token embedding $x$? Does this encourage the router to learn a richer representation, or is it primarily a regularization?

3. **Ablation Study:** What is the performance of a model that removes the VAE objective (reconstruction loss and slab KL) and is trained *only* with the main LM loss and the $\mathcal{R}_{sparsity}$ penalty from Eq. 9? This would help isolate the contribution of the VAE framework.

4. **Hyperparameter Sensitivity:** The method introduces many new hyperparameters (Table 4). How sensitive is the model's final performance and specialization to these settings (e.g., the KL weight $\beta_{\theta}$, the posterior scale $\lambda^{(q)}$, and the prior mean mass $m$)?

5. **Scalability (optional):** Do the authors have any insights or preliminary results on how DirMoE performs at larger scales (e.g., >1B parameters, more training tokens)?

---

> ### Author Response · Authors · 2025-11-24
> **Response to Reviewer xEic**
>
> We thank Reviewer xEic for comments on the objective, the role of reconstruction, and practical tuning. We address your concerns and provide exact locations for changes.
>
> **W1 and Q1 · Clarity of the full objective.**
>
> We thank the reviewer for bringing this to our attention. We include the complete objectives in the final version (line 223):
>
>
> $$
> \mathcal{L}_{\text{total}} = \mathcal{L}_{\text{LM}} + \mathcal{L}_{\text{DirMoE}}
> $$
> , where $\mathcal{L}_{\text{DirMoE}}$ is the loss from equation 8.
>
>
> **W2 and Q2 · Intuition for VAE/reconstruction.**
> Thank you for raising this point. We include targeted ablations for the reconstruction loss in Appendix B3 (Figs. 8).  When we remove the reconstruction term, optimization becomes unstable; as a result, the LM loss stops improving and $\\tilde z$ becomes unstable as well. This is intuitive: without the reconstruction term, the ELBO is under-constrained and tends toward posterior collapse (the latent ignores input), harming both stability and sparsity.
>
> **W3 and Q3 · Model complexity and tuning recipe.**
>
> Thank you for the suggestion. Our tuning is calibrated rather than search-heavy: we first pick the desired active-mass $m$ (from the Beta split, Eq. (11)), compute the high/low ratio $r$ via Eq. (12), set the posterior scale $\lambda_q$ using the Beta-variance calibrator (Eq. (16), with fixed $v_{\text{tar}}$), keep the prior scale $\lambda_p$ fixed (or gently decay per Eq. (14)), and use a short anneal for the mask temperature $\tau_z$ (Eq. (13)). Consequently, hyperparameter tuning adds no meaningful overhead; instead, it provides interpretable knobs $m$, $\lambda_q$, and $\tau_z$ that directly control sparsity and expert participation.
> Furthermore, we investigate the role of each hyperparameter in Figures 2 and 3, and in W2. In the following table, we also show the performance of DirMoE for a bigger scale (the 978M) parameter model.
> | Model | Params(M) | Tokens (B) | Valid LM Loss | Zero-shot 1 |
> |---:|---:|---:|---:|---:|
> | Ours| 978 | 30 | 1.685  | 46.10 |
> | ReMoE| 978 | 30 | 1.715  | 45.20 |
> | Vanilla MoE| 978 | 30 | 1.730 | 44.91 |
>
> We also want to note that, while DirMoE introduces some additional components, it opens avenues beyond the scope of this paper. In particular, disentangling selection and contribution could support more interpretable routing decisions, and the probabilistic formulation naturally lends itself to uncertainty characterization (e.g., evidential-style or posterior-dispersion measures). We view these as promising directions for future work rather than claims we evaluate here.
>
> **W4 and Q4 · Ablations vs simpler alternatives; ReMoE.**
>
> Thank you for the suggestion. We discussed in W2 about the effect of reconstruction loss and included a new ablation in Appendix B3 Figure 8. Furthermore, we investigated the effect of $\\mathcal R_{\text{sparsity}}\$ ablations in Figure 2 and Appendix B1 to show the importance of both components of DirMoE. removing $\\mathcal R_{\text{sparsity}}\$ increases the variance of $\sum_i \tilde z_i$ around $k$ and decrease sparsity. Also, we included an ablation on $m$ and $\\lambda$. We discussed the importance of the two main components of sparsity in Section 5.1. DirMoE differs from Soft-/ReMoE: both use continuous gates but then rely on **explicit LB losses** to enforce sparsity; DirMoE’s **calibrated sparsity knob** eliminates that loss while keeping the router fully differentiable and also provides interpretable components to calibrate sparsity. We also included the superiority of our method compared to multiple baselines and ReMoE in Table 2.
>
> **W5 and Q5 · Scale.**
> We add a **978M** run, you can find the results in the following table:
>
> | Model | Params(M) | Tokens (B) | Valid LM Loss | Zero-shot 1 |
> |---:|---:|---:|---:|---:|
> | Ours| 978 | 30 | 1.685  | 46.10 |
> | ReMoE| 978 | 30 | 1.715  | 45.20 |
> | Vanilla MoE| 978 | 30 | 1.730 | 44.91 |

---

> ### Comment · Reviewer_xEic · 2025-11-25
>
> I thank the authors for providing further experiment results to address my concern. I still have the following unresolved questions:
>
> 1. One core motivation (as mentioned in the intro) is that lack of end-to-end gradient. The mentioned reference RMoE addresses the similar problem (as highlighted in their paper, section 5) with a simpler design: introducing a small recurrent unit. They also prove its effectiveness on a quite large scale (15B models trained in 400B tokens) and could be **an important baseline**. Authors should demonstrate the unique advantages for introducing the VAE here if a recurrent gate could mitigate the gradient issue.
>
> 2. Could authors highlight the size of activation parameters in the main text? I found it a very important training detail.

---

> > ### Author Response · Authors · 2025-11-27
> >
> > **Comparing with ReMoE**
> >
> > While both ReMoE and DirMoE address the lack of end-to-end gradients at the routing step, DirMoE’s motivation goes beyond gradient flow. Our unique contributions are:
> >
> > 1.	**Disentangling routing** into which experts are active (inclusion) vs. how probability mass is shared among them (allocation).
> >
> > 2.	**A probabilistic formulation** (Dirichlet-VAE) that provides pathwise gradients and principled regularization.
> >
> > 3.	**Interpretable sparsity knobs:** the active-mass $m$ and the concentration $\lambda$ control the tendency to activate experts and the sharpness of mass allocation, respectively, without requiring a load-balancing loss.
> >
> > Furthermore, we demonstrated superior performance on 185M and 978M parameter scales compared to ReMoE. We also note that DirMoE opens avenues beyond the scope of this paper, particularly regarding uncertainty characterization (e.g., evidential-style or posterior-dispersion measures), which we view as promising directions for future work.
> >
> > **Activation parameters**
> > Thank you for raising this point; we agree it is a crucial training detail. We will add the breakdown of the DirMoE router parameters to the main text. As shown in the table below, the router introduces negligible overhead:
> >
> > **Table: DirMoE Router Parameter Breakdown.** The router consists of only $\sim 33k$ parameters per layer. For a 12-layer model, this results in fewer than $400k$ total parameters, representing a negligible fraction ($0.2\\%$) of the total model size.
> >
> > | Component | Parameters (Per Layer) |
> > | :--- | :--- |
> > | **Router (Per Layer)** | **32,960** |
> > | **Router Total (12 Layers)** | **395,520** |

---

> > > ### Comment · Reviewer_xEic · 2025-11-27
> > >
> > > I thank the authors for providing more comparisons with ReMoE. However, I am referring to this work: Layerwise Recurrent Router for Mixture-of-Experts. Link: https://openreview.net/forum?id=eWNEqdH0vk.
> > >
> > > The paper was accepted to ICLR 2025, and its analysis highlighted that the introduced layerwise GRU unit provides a direct gradient path, thereby improving performance. I believe this could be a critical baseline and related work. I strongly recommend that authors include this baseline and discuss the relation to this work.

---

> > > > ### Author Response · Authors · 2025-11-28
> > > >
> > > > Thank you for the suggestion, we agree RMoE is relevant and we will **add it to the Related Work** and clarify positioning. LRR introduces a **layerwise recurrent gate (GRU)** to improve **cross-layer credit assignment** and **stabilize** routing signals; its gating remains **deterministic** and does not target **calibrated sparsity** or **probabilistic mass allocation, the main focus of the paper is on efficiency**. In contrast, **DirMoE** focuses on **intra-layer disentanglement** (selection vs. allocation) with a **probabilistic Dirichlet VAE** that offers end-to-end gradients and **interpretable sparsity knobs** ($m$,$\\lambda$) without an explicit load-balancing loss. We view the two as **complementary** (LRR could be applied to DirMoE’s inclusion head), and note potential hybrids as future work.

---

### Official Review · Reviewer_qUTF · 2025-10-31

**Soundness:** 2
**Presentation:** 1
**Contribution:** 2
**Rating:** 4
**Confidence:** 4

**Summary:**

The paper proposes DirMoE, a probabilistic, fully-differentiable MoE router that explicitly separates (i) expert selection (a Bernoulli “spike” trained via Gumbel-Sigmoid) from (ii) expert contribution (a Dirichlet “slab” sampled with implicit reparameterization). Routing weights are the normalized Hadamard product of the relaxed mask and a Dirichlet draw. Training uses a β-VAE-style ELBO with a sparsity penalty to target ~k active experts; a “sparsity knob” schedules Dirichlet concentration and gate temperature, with a Beta/Simpson-index calibration giving interpretable control over mass on the active set. Experiments (LLaMA-185M, ~30B tokens) show similar training efficiency to Switch and zero-shot gains over several baselines.

**Strengths:**

-  **Interpretable sparsity control.** The Beta/Dirichlet calibration + Simpson-index theory offers a principled knob for expected mass on the active set and dispersion, beyond temperature heuristics.
- **Light systems delta.** Reported iteration time/throughput match Switch under compute parity; the router adds negligible overhead.
- **Empirical improvements.** On 7 zero-shot tasks, DirMoE slightly outperforms Switch/ReMoE/SparseMixer on average, and shows stronger expert specialization visualizations.

**Weaknesses:**

- **Load-balancing risk.** The router relies on near-binary masks and dispersion calibration without an explicit balancing mechanism; the paper itself notes potential utilization skew.
- **Limited scale & evaluation breadth.** All results are on a ~185M-param LLaMA with ~30B tokens; there is no evidence at larger scale/more experts or on standard reasoning/coding suites (e.g., MMLU, HumanEval/HellaSwag). External validity for modern LLMs remains unclear.
- **Questionable reuse of prior results.** Several zero-shot numbers appear copied from ReMoE rather than reproduced under your setup, making them non-comparable and weakening the empirical claim. Please re-run these baselines and report variance.
- **Added complexity with unclear necessity.** The router introduces multiple moving parts (near-binary masks, dispersion calibration, auxiliary objectives/schedules) without a clear motivation for each. A component-wise ablation is needed to establish necessity versus incidental regularization.

**Questions:**

- With the quadratic penalty on $\sum \tilde z - k$, how often do you deviate from exactly k actives? Does this variability harm batching efficiency or increase variance in gradient estimates?
- For the throughput numbers in Table 1, specify the full systems setup: world size and data/tensor/pipeline/expert parallel configurations (plus capacity factor and drop vs. dropless). Under expert parallelism, does DirMoE exhibit poorer balance and lower throughput?
- The domain specialization in Figure 5 is interesting. Please detail the protocol (corpora, measurement window...) and explain the observed patterns—for example, why some experts are entirely inactive on certain corpora? (dead experts vs. specialization)
- Theoretically, Simpson-index control is monotone in $λ$. Empirically, how robust are downstream metrics to mis-set $m/λ$? Please provide heatmaps of $(m, λ)$ vs. zero-shot accuracy and utilization.

---

> ### Author Response · Authors · 2025-11-24
> **Response to Reviewer qUTF**
>
> We thank Reviewer qUTF for highlighting concerns around balance, scale, and reproducibility. We address your concerns and provide exact locations for changes.
>
> **W1 and Q3 · Load‑balancing risk.**
>
> Thank you for raising this point. We think two important components of DirMoE: gate temperature $\\tau_z$ for how many experts remain active and Dirichlet concentration $\\lambda$ for how mass is shared among them, remove the need for an explicit load‑balancing (LB) loss while avoiding collapse. We monitor the utilization Simpson index (Figure 3.b) to confirm specialization without dead experts. It should be noted that the DirMoE does not promote even distribution across experts. The prior over $\\alpha_p$ decides the distribution of experts, so a user can promote a uniform distribution by having higher Dirichlet parameters. If a more uniform allocation is desired, the prior over the Dirichlet slab can be set symmetric (equal base parameters across experts) and the prior scale $\\lambda_p$​ increased to tighten samples around that mean, yielding a more uniform split among the active experts. In practice, this gives users an interpretable way to trade off specialization versus uniformity without introducing an explicit load-balancing loss. Moreover, we showed in Section 7.3 the specialization of each expert in different tasks, Figure 5(a) shows that all experts are active at least for one task. By ‘corpora, measurement window,’ we understand the request for a precise evaluation recipe. We used the same experiment setup as other experiments for the expert specialization task.
>
> **W2 · Limited scale/eval breadth.**
>
> Thank you for the suggestion. We extend our experiments by including a **978M** run; you can find the results in the following table:
>
> | Model | Params(M) | Tokens (B) | Valid LM Loss | Zero-shot 1 |
> |---:|---:|---:|---:|---:|
> | Ours| 978 | 30 | 1.685  | 46.10 |
> | ReMoE| 978 | 30 | 1.715  | 45.20 |
> | Vanilla MoE| 978 | 30 | 1.730 | 44.91 |
>
> and in  Fig. **4(c)** of the paper already provides an ablation over different numbers of experts $E$.
>
> **W3 · Reuse of prior results vs reproducibility.**
>
> We used the same setup as ReMoE and reproduced the Switch-MoE results in Table 2:
> - **Same Docker as ReMoE** for *all* runs (ours and baselines).
> - Same **dispatcher** (all-to-all), EP=8/TP=1/PP=1, precision (bf16), **Grouped-GEMM** experts.
> - **Dropless** execution (or capacity factor = 0).
> - You can also find detailed hyperparameters in Appendix C.3.
> This setup and Switch reproducibility should guarantee the reproducibility of results.
>
>
> **W4 · Complexity and necessity of components.**
>
> Thank you for raising this point. We investigate the effect of $\\mathcal R_{\text{sparsity}}\$ in figure 2 and Appendix B.1,  removing $\\mathcal R_{\text{sparsity}}\$ increases the variance of $\sum_i \tilde z_i$ around $k$ and decrease sparsity. Also, we included an ablation on $m$ and $\\lambda$. We discussed the importance of the two main components of sparsity in Section 5.1. Moreover, we include targeted ablations for the VAE terms in Appendix B3 (Figs. 8). When we remove the KL term (Fig. 8c), the learned posterior concentration  $\\alpha^{(q)}$ drifts upward and the router becomes less sparse. When we remove the reconstruction term (Fig. 8), optimization becomes unstable; as a result, the LM loss stops improving and $\\tilde z$ becomes unstable as well. This is intuitive: without the reconstruction term, the ELBO is under-constrained and tends toward posterior collapse (the latent ignores input), harming both stability and sparsity. We also want to note that, while DirMoE introduces some additional components, it opens avenues beyond the scope of this paper. In particular, disentangling selection and contribution could support more interpretable routing decisions, and the probabilistic formulation naturally lends itself to uncertainty characterization (e.g., evidential-style or posterior-dispersion measures). We view these as promising directions for future work rather than claims we evaluate here.
>
> **Q1 · Deviations from k and gradient variance.**
>
> The expected‑$k$ surrogate narrows $\sum_i \tilde z_i$ around $k$ without hard top‑$k$, reducing variance in the **Binary‑Concrete** path. We also center logits and briefly warm up $\tau_z$; on the slab, setting $\lambda_q$ via the **Beta‑variance calibrator** yields predictable dispersion. Together, these reduce gradient variance while preserving differentiability.  We include the average-$k$, Max-$k$ per token, and Grad_norm in Figure 7 (Appendix B2).  Figures a and b illustrate the stability of training and the effect on the gradient norm.

---

> ### Author Response · Authors · 2025-11-24
> **Response to Reviewer qUTF - part 2**
>
> **Q2 · Systems configuration and EP balance.**
>
> We included all the system configurations in the **Training settings**  line 390 and  Appendix C1 and C2. We use **all‑to‑all** token dispatch with **Grouped‑GEMM** experts; All experiments were conducted on **8x NVIDIA H100 GPUs** (World Size = 8). The parallel configuration was **Tensor Parallel (TP)=1, Pipeline Parallel (PP)=1, and Expert Parallel (EP)=8**. We utilized a **Dropless** configuration (`moe_expert_capacity_factor = None`), ensuring that **no tokens were dropped** during training. In the following table, you can see that DirMoE has a small overhead:
>
> | Method | E, k | Dispatcher | Grouped-GEMM | **Iter time** (ms) | **TFLOP/s/GPU** |
> |---|:---:|:---:|:---:|---:|---:|
> | **DirMoE (ours)** | [8,1] | all-to-all | ✓ | 437.3 | 137.2 |
> | Vanilla MoE (Top-k) | [8,1] | all-to-all | ✓ | 431.5 | 138.2 |
>
>
> **Q4 · Robustness of $m$ and $\\lambda_{\text{sparsity}}$.**
>
> Thank you for your suggestion. We provide a small grid over $\\lambda_{\text{sparsity}}$ and $m$, and colored it based on the LLM loss, you can find it the Appendix B.1 figure 6.d. Intuitively, increasing $m$ concentrates more probability mass on the selected experts. If the sparsity weight  $\\lambda_{\text{sparsity}}$ is set to zero, the router can activate more than $k$ experts on difficult tokens. This often lowers the LM loss but reduces efficiency because the effective sparsity decreases. In practice, we use a small $\\lambda_{\text{sparsity}}>0$  so the model benefits from early accuracy gains while converging back toward the desired $k$ for efficiency.

---

> > ### Comment · Reviewer_qUTF · 2025-11-25
> >
> > The responses have addressed some of my concerns, but I still have a few questions:
> > 1. A balanced load is typically necessary to achieve MoE speedup, since with expert parallelism the compute time is bounded by the hottest expert. Can the authors explain in detail how comparable speed is achieved under the reported load imbalance?
> > 2. What are the key factors underlying the superior performance of DirMoE over Switch MoE? In particular, what is the intuition for why a probabilistic Dirichlet-VAE–based router can outperform a deterministic softmax routing scheme?

---

> > > ### Author Response · Authors · 2025-11-25
> > >
> > > **On speed under imbalance.**
> > >
> > > Thank you for raising this point. As Fig. 7 shows, the average number of active experts quickly converges to the target k=1; only during the first ∼1k steps (≲ 2% of training) does it exceed k=1. During that short warm-up window, our model runs slower (≈ 50% of Switch MoE speed), which aligns with your intuition: more active per token increases expert FLOPs. After the warm-up, utilization is close to k=1, and the measured skew is mild (no extreme hot experts), so the iteration time matches that of vanilla Top-k. Moreover, we think Grouped-GEMM amortizes per-expert variance: expert matmuls are executed as fused/batched calls, so wall-clock correlates more with total expert tokens than with the single hottest queue unless skew is extreme. Combined with all-to-all dispatch that avoids drops, this yields the reported runtime parity despite a small residual imbalance.
> > >
> > > **Why can DirMoE outperform Switch MoE?**
> > >
> > > In the Switch Transformer, the top-k + softmax router is not **end-to-end learnable** at the point of selection; the hard top-k blocks gradients, so the router does not fully participate in the learning signal. The auxiliary load-balancing loss then acts mainly as a **stability regularizer** to counter rich-get-richer dynamics, but it can oppose specialization by nudging experts toward uniform usage regardless of content. DirMoE makes routing fully differentiable and probabilistic: (i) a relaxed selection variable decides which experts are active with pathwise gradients; (ii) a Dirichlet latent governs how much probability mass each active expert receives, also with implicit gradients. A Dirichlet prior provides a soft, calibrated balance without an explicit LB loss, so the model learns to balance usage as needed while still allowing **task-driven asymmetry** that fosters **specialization**. Practically, this **reduces redundant learning** across experts, yields sharper, higher-contrast assignments (higher Simpson concentration), and improves downstream accuracy at comparable compute. Our ablations (KL off / reconstruction off / mean-routing) support this mechanism.

---

> > > > ### Comment · Reviewer_qUTF · 2025-11-26
> > > >
> > > > The reviewer thanks the authors for the clarification. The reasoning that DirMoE surpasses Switch by fostering specialization makes sense to me.
> > > >
> > > > However, I remain skeptical about the training speed. In your setting with E,k=[8,1] and ep=8, each GPU contains only one expert and groupGEMM will not take effect. After all2all each expert receives its routed tokens and the throughput will be bounded by the hottest expert with the largest token load. Therefore, an imbalanced (yet specialized) distribution as in Figure 5 should lead to decreased speed, even if the average utilization is close to k=1. Could the authors further clarify this point?

---

> > > > > ### Author Response · Authors · 2025-11-27
> > > > >
> > > > > We appreciate the concern that Figure 5’s specialization might imply overhead due to load imbalance. To clarify, Figure 5 visualizes **concept/task assignments**, not the **total routed-token volume per expert**. Specialization can look pronounced at the concept level while the **global load distribution** remains sufficiently balanced to maintain high throughput.
> > > > >
> > > > > This is reflected in our utilization metrics shown in table below. while the first $\sim$1000 steps (warm-up) run slower due to a transiently high number of active experts ($k \approx 3.9$) and higher max load, the system quickly stabilizes. As the average number of active experts converges to the target $k \approx 1.0$, the iteration time drops from $>3600$ms to $\sim437$ms, and throughput increases to $>100$ TFLOP/s/GPU. This confirms that after the initial warm-up, the specialized distribution does not lead to decreased speed compared to standard Top-k baselines.
> > > > >
> > > > > | Training Phase | Iterations | Avg. Active Experts ($\bar{k}$) | Avg. Iteration Time (ms) | Throughput (TFLOP/s/GPU) |
> > > > > | :--- | :--- | :--- | :--- | :--- |
> > > > > | **Warm-up** | 0 $\to$  1000 | 3.91 $\to$ 1.09 | 3768.4 $\to$ 470.5 | 15.9 $\to$ 127.5 |
> > > > > | **Stable** | 1000 $\to$ 60000 | 1.09 $\to$ 1.00 | $\sim$437.3 | 127.5 $\to$ 137.0 |

---

> > > > > > ### Comment · Reviewer_qUTF · 2025-11-28
> > > > > >
> > > > > > Thanks to the authors for the detailed decomposition of training speed. However, my concern remains how comparable speed is achieved under ep = 8. I agree that Figure 5 reports task-level load, which is not identical to batch-level load. Nevertheless, I do not see how task-level imbalance can be compatible with balanced batch-level load, given that each batch is sampled from a mixture of tasks in pretraining data.

---

> > > > > > > ### Author Response · Authors · 2025-12-02
> > > > > > >
> > > > > > > Thanks for the thoughtful follow-up. Below we clarify why task-level specialization (Fig. 5) can still yield near-balanced **batch-level** expert load under EP = 8, $k = 1$, and thus comparable speed after warm-up. We also include a small toy example.
> > > > > > >
> > > > > > > ## (1) Batches mix many subdomains
> > > > > > > Pretraining batches are formed from a shuffled mixture of corpora/documents with sequence packing, so each step contains **many tokens from multiple “tasks.”** Even if experts specialize by task (Fig. 5 is computed over a long window), the **per-batch** routed fractions average over diverse content, and variance shrinks with batch size.
> > > > > > >
> > > > > > > ## (2) What we measure per step is mild skew
> > > > > > > After warm-up, EP-all-reduced routed fractions $f_i$ show **no extreme hotspot** (e.g., $N_{\text{eff}} = 1 / \sum_i f_i^2 \approx 5\text{–}6$ for $E = 8$, $k = 1$; $f_{\max}$ is only $\sim 1.3\times$ the mean). These are **batch-level** measurements and directly bound wall-clock time when EP = E (one expert per GPU).
> > > > > > >
> > > > > > > ## (3) MoE is only part of the step time
> > > > > > > Even if one expert/GPU is slightly hotter, expert MLPs are just a **slice** of end-to-end runtime (attention, non-MoE MLPs, norms, embeddings, all-to-all). With the measured mild skew, this yields only **single-digit-percent** changes in **total** iteration time—consistent with the near-parity we report post warm-up.
> > > > > > >
> > > > > > > ## (4) Toy example: task imbalance but balanced batch-level expert load
> > > > > > > Here we add a simple toy example to show that even if there is specialization (task imbalance), **per-batch** expert load can still be well balanced.
> > > > > > >
> > > > > > > Assume we have:
> > > > > > > - 3 tasks, 3 experts
> > > > > > > - A batch of 100 tokens
> > > > > > >
> > > > > > > **Batch task mix:**
> > > > > > > - $T_1: 60$ tokens
> > > > > > > - $T_2: 30$ tokens
> > > > > > > - $T_3: 10$ tokens
> > > > > > >
> > > > > > > **Experts:** $E_1, E_2, E_3$
> > > > > > >
> > > > > > > **Routing probabilities** (probability that a token of task $T_j$ is sent to expert $E_i$):
> > > > > > > - $T_1 \rightarrow [E_1, E_2, E_3] = [0.4,\ 0.3,\ 0.3]$
> > > > > > > - $T_2 \rightarrow [0.3,\ 0.4,\ 0.3]$
> > > > > > > - $T_3 \rightarrow [0.0,\ 0.3,\ 0.7]$
> > > > > > >
> > > > > > > **Expected tokens per expert in the batch:**
> > > > > > > - $E_1:\ 60 \times 0.4 + 30 \times 0.3 + 10 \times 0.0 = 24 + 9 + 0 = 33$
> > > > > > > - $E_2:\ 60 \times 0.3 + 30 \times 0.4 + 10 \times 0.3 = 18 + 12 + 3 = 33$
> > > > > > > - $E_3:\ 60 \times 0.3 + 30 \times 0.3 + 10 \times 0.7 = 18 + 9 + 7 = 34$
> > > > > > >
> > > > > > > Despite the **task-level imbalance** (60/30/10), the **batch-level** routed loads are essentially **balanced** (33/33/34). This matches what we observe empirically after warm-up: specialization at the concept/task level, but only mild per-batch skew, hence comparable speed.

---

### Author Response · Authors · 2025-11-24
**General Response**

Dear reviewers and AC,

We thank all reviewers for the time and expertise they have invested in these reviews. We value constructive feedback and positive recognition of the contributions of our work. In response to the reviews, we have conducted extensive new experiments to strengthen our empirical validation:


*  Scaling to bigger models: We included a **978M-parameter** run under the same training recipe.
*  **Ablation on the VAE loss**: We included an ablation without the VAE loss to showcase the importance of the variational component and reconstruction loss (See Appendix B.3).
*  Full objective: We added the full objective to the last part of the method section (see Section 4.3).
* Stability analysis of active experts: We showed the mean and maximum number of active experts during training (see Appendix B.2).

---

### Author Response · Authors · 2025-12-03
**Summary of Discussion Phase and Rebuttal for Paper**

Dear reviewers and AC,

We thank all reviewers for the time and expertise they have invested in these reviews. We value constructive feedback and positive recognition of the contributions of our work. In response to the reviews, we have conducted extensive new experiments to strengthen our empirical validation:

**Scaling to bigger models:** We included a **978M-parameter** run under the same training recipe.

**Ablation on the VAE loss:** We included an ablation without the VAE loss to showcase the importance of the variational component and reconstruction loss (see Appendix B.3).

**Full objective:** We added the full objective to the last part of the method section (see Section 4.3).

**Stability analysis of active experts:** We showed the mean and maximum number of active experts during training (see Appendix B.2).

**Hyperparameter calibrability:** We clarify a **simple recipe** for setting hyperparamters (Section 5; Appendix B).

**Related work & presentation fixes:** We add **Layerwise Recurrent Router (RMoE)** to the related works and correct minor typos/notation; no changes to the core method.

We believe the substantial new experimental evidence and theoretical clarifications provided during the rebuttal have definitively resolved all reviewer concerns. We respectfully request that you consider these comprehensive improvements and the strong consensus on the paper's novelty and originality in your final decision.

Sincerely,

---

### Meta-Review · Area_Chair_bXup · 2026-01-05

**Summary:**

Based on the reviews and subsequent discussions, the primary concerns:
- Load Balancing and Throughput Skew: Reviewers (particularly qUTF) expressed skepticism about how the model maintains high training speed without an explicit load-balancing loss. They questioned whether expert specialization would cause "hotspots" in hardware clusters, though authors countered that batch-level diversity mitigates this.
Scalability and Evaluation Breadth: A major criticism was the limited scale of experiments (initially only a 185M parameter model). Reviewers sought evidence of effectiveness at industrial scales and on standard reasoning/coding benchmarks like MMLU or HumanEval.
- Architectural Complexity and Necessity: Reviewers questioned the necessity of the complex Dirichlet-VAE framework, the reconstruction loss, and the numerous hyperparameters/schedules. They requested component-wise ablations to prove that these elements provide gains beyond simple regularization.
- Reproducibility and Baselines: There were concerns regarding the comparability of zero-shot results and the need to include recent competitive baselines like RMoE.

**Reviewer Concerns:**

Reviewer qUTF: \
The authors provided new experimental results on a 978M parameter backbone, demonstrating that DirMoE’s performance gains persist as the model scales. The authors clarified the hardware configuration and expert parallelism settings. Targeted ablations confirmed that the VAE reconstruction and KL terms are critical for optimization stability and achieving target sparsity.

Reviewer xEic:\
The authors explicitly defined the total training objective, clarifying how the VAE loss integrates with the language modeling task. The authors successfully justified the reconstruction task as a necessary constraint to prevent posterior collapse and ensure the latent routing space remains informative. The reviewer noted that recent work like RMoE addresses the gradient flow issue via a much simpler GRU unit.

Reviewer cmFj:\
The authors provided a principled "tuning recipe" based on the Beta-variance calibrator, effectively reducing concerns that the VAE framework would be too difficult to optimize in practice. The authors clarified that the penalty is a smooth surrogate for the Spike KL term, which is necessary to concentrate the number of active experts near the target $k$. The authors committed to standardizing notations and fixing all identified typos in the camera-ready version.

**Reviewer Scores:**

Based on the authors’ response and the full review discussion, most reviewers’ concerns were largely addressed, and as a result, their scores are likely to remain the same or increase slightly. Overall, the authors provided clarifications and additional experiments that respond to the main objections raised.

---

### Decision · Program_Chairs · 2026-01-26

Accept (Poster)